# Adaptive Priors from Learning Trajectories for Function-Space Bayesian Neural Networks

## Abstract

Tractable Function-space Variational Inference (T-FVI) provides a way to estimate the function-space Kullback-Leibler (KL) divergence between a random prior function and its posterior. This allows the optimization of the function-space KL divergence via Stochastic Gradient Descent (SGD) and thus simplifies the training of function-space Bayesian Neural Networks (BNNs). However, function-space BNNs on high-dimensional datasets typically require deep neural networks (DNN) with numerous parameters, and thus defining suitable function-space priors remains challenging. For instance, the Gaussian Process (GP) prior suffers from scalability issues, and DNNs do not provide a clear way to set appropriate weight parameters to achieve meaningful function-space priors. To address this issue, we propose an explicit form of function-space priors that can be easily integrated into widely-used DNN architectures, while adaptively incorporating different levels of uncertainty based on the function's inputs. To achieve this, we consider DNNs as Bayesian last-layer models to obtain the explicit mean and variance functions of our prior. The parameters of these explicit functions are determined using the weight statistics over the learning trajectory. Our empirical experiments show improved uncertainty estimation in image classification, transfer learning, and UCI regression tasks.

## 1 Introduction

Function-space Bayesian neural networks (BNNs) (Sun et al., 2019) have gained significant attention within the Bayesian deep learning community, primarily due to their fundamental goal of assigning prior distributions to the outputs of neural networks directly. Training these BNNs can generally be conducted by optimizing the function-space Evidence lower bound (ELBO) consisting of the expected likelihood and the function-space KL divergence between a random prior function and its posterior function (Sun et al., 2019). The recent tractable function-space variational inference (T-FVI) (Rudner et al., 2022) presents the closed form of function-space KL divergence using linearized NNs, and thus facilities the optimization of the training objective via Stochastic Gradient Descent (SGD). However, recent function-space BNNs use DNN architecture using many parameters to model high-dimensional dataset and thus raises a challenge in setting the suitable priors for the function-space BNNs.

Gaussian process (GP) has been a representative function-space (Rasmussen, 2004). This prior has been used for the small-sized BNNs conducting the regression or low-dimensional classification (Flam-Shepherd et al., 2017; Tran et al., 2022). However, GP prior for modeling high dimensional datasets has scalable issues in training the kernel hyperparameters. Thus, GP prior is rarely used as the function-space prior for the commonly-used DNN architectures, such as ResNet.

Alternatively, mapping weight-space prior to function-space prior through the linearized NNs can be considered for setting the prior of such DNN architectures (Rudner et al., 2022). However, since the derived function-space prior might incorporate largely different prior into the model's output according to the assigned weight-space prior, it requires to carefully set the weight-space prior and thus limits to practically use of this approach. Additionally, this approach practically restricts the randomness to the last layer for Jacobian computation because it requires a large amount of GPU memory to compute the large-sized Jacobian matrix of NN for each input. This practical usage might reduce the flexibility in the resulting BNNs.

Furthermore, function-space VI requires the external dataset for computing function-space KL divergence because this KL term measures the distance between two random functions defined in an infinite-dimensional space (Sun et al., 2019; Rudner et al., 2022). Employing the well-curated external datasets can enhance the model's uncertainty estimation capabilities (Antorán et al., 2023; Lopez et al., 2023). On the other hand, the arbitrary-chosen external dataset, without considering its relationship with the training set, may adversely impact training.

In this work, we propose an explicit form of function-space prior that can be easily used for the widely-used DNN architectures, and adaptively introduce different levels of uncertainty based on the function's inputs. To this end, we consider DNNs as Bayesian last-layer models, yielding a closed form of the function-space prior. Then, we devise the explicit mean function and variance functions of our prior to adaptively produce higher uncertainty for each function's output, similarly to GP. We set the parameters of these explicit mean and variance functions by using the weight statistics over the learning trajectory. Additionally, based on the property of designed prior, we propose an adversarial context feature that can be used for computing the function-space KL divergence without relying on external datasets. We expect this context feature to impose additional uncertainty into the model's output on potential Out-of-distribution (OOD) inputs. Our implementation is available here. We summarize our contribution as follows:

- We propose an explicit function-space prior that can be easily used for the common DNN architectures as well as adaptively incorporate higher uncertainties for each function's input.
- We propose a context feature to compute the function-space KL without using external datasets.
- We showcase the effectiveness of our approach across diverse benchmark tasks. Notably, our prior is more effective in experiments involving large-scale models like vision transformers (Dosovitskiy et al., 2021).

## 2  BACKGROUND.

**Settings and Notations.**  In this work, we focus on Bayesian neural network (BNN) for supervised learning task. Let $\mathcal{X} \subset \mathbb{R}^D$ and $\mathcal{Y} \subset \mathbb{R}^Q$ be the space of inputs and outputs, respectively. Let $f : \mathcal{X} \times \mathbb{R}^P \longrightarrow \mathcal{Y}$ be a BNN that takes the input $x \in \mathcal{X}$ and the random weight parameters $\theta \in \mathbb{R}^P$, following prior distribution $p(\theta)$, and produces the random output $f(x, \theta) \in \mathcal{Y}$. For parameter representation, we notate a vector form $\theta$ and its matrix form $\Theta$, i.e, $\theta = \text{vec}(\Theta)$ and $\Theta = \text{vec}^{-1}(\theta)$ . When it's evident, we omit the parameter $\theta$ and write $f(x)$ instead.

For the notation of vector and matrix, we denote a matrix $A \in R^{N \times M}$ using uppercase letter and its $k$-th row $[A]_{k,:}$ and $j$-th column $[A]_{:,j}$. We denote a vector $x \in R^D$ using lowercase letter and its $i$-th entry $[x]_i$. We notate weighted norm $\|x\|_w^2 = x^\top \text{diag}(w) x$ for a weight vector $w \in R^D$.

**Function Space Variational Inference for BNNs.**  Function space BNNs introduce the prior distribution on the output of the Deep Neural networks (DNN) to incorporate the inductive bias into the model. Due to the intractability of the posterior distribution, the function-space BNNs are generally trained with the function space variational inference (VI). Given a dataset $\mathcal{D} = \{(x_n, y_n)\}_{n=1}^N$ with input $x_n \in \mathcal{X}$ and $y_n \in \mathcal{Y}$, let $p(f)$ be the prior distribution of the model output $f$ and $q(f)$ be its variational posterior distribution with a variational parameter $\phi$, where we omit $\phi$ from the notation. The variational parameter $\phi$ is then optimized by maximizing the Evidence Lower Bound (ELBO):

$$\mathcal{L}_{\text{fvi}}(\phi) = \mathrm{E}_{q(f)} \left[ \sum_{n=1}^N \log p(y_n | f(x_n)) \right] - \lambda \, \text{KL}(q(f) \| p(f)), \tag{1}$$

where $\lambda$[1] is the hyperparameter controlling the regularization effect from the KL divergence. As both $p(f)$ and $q(f)$ are in principle stochastic processes, the KL divergence in Eq. (1) is defined as,

$$\text{KL}(q(f) \| p(f)) = \sup_{X_{\text{ctx}} \subseteq \mathcal{X}^m} \text{KL}(q(f(X_{\text{ctx}})) \| p(f(X_{\text{ctx}}))), \tag{2}$$

(Sun et al., 2019), where a *context set* $X_{\text{ctx}} \subseteq \mathcal{X}^m$ for some $m \in \mathbb{N}$ denotes a finite number of dataset and $f(X_{\text{ctx}}) := (f(x))_{x \in X_{\text{ctx}}}$ and similar for $q(X_{\text{ctx}})$. In practice, evaluating the supremum is

---

[1]Setting $\lambda < 1$ is equivalent to optimizing a *tempered posterior distribution*, which usually performs better than a vanilla Bayes posterior. This phenomenon is well known as the cold posterior effect (Wenzel et al., 2020).

intractable, and it is typically approximated with a heuristically chosen context set $X_{\text{ctx}}$. A naïve way is to sample $X_{\text{ctx}}$ as a random subset from the training set. (Rudner et al., 2022) suggests utilizing an external dataset that closely aligns with the original training set but is not identical. Even with this approximation, the KL divergence evaluated on the context set $\text{KL}\big(q(f(X_{\text{ctx}}))\|p(f(X_{\text{ctx}}))\big)$ in Eq. (2) may not admit a closed-form expression. Optimizing this KL term needs an additional technique of the gradient estimation (Sun et al., 2019; Shi et al., 2018).

**Tractable Function-Space Variational Inference for BNN.** Rather than directly eliciting a prior distribution $p(f)$, one can initially choose a weight-space prior $p(\theta)$ and then define the function-space prior $p(f(x, \theta))$ as an induced distribution $p(f(x, \theta)) := \int_{\mathbb{R}^P} \delta_\theta(\theta') f(x, \theta') p(\theta') d\theta'$. Based on this prior, (Rudner et al., 2022) proposed a tractable function-space variational inference method using the linearized BNNs with respect to the weight parameters to make the computation of the KL term in Eq. (2) tractable. Specifically, for the prior distribution of the weight parameters $p(\theta) = \mathcal{N}(\theta; \mu, \text{diag}(\sigma^2))$, the linearized BNN $f_{\text{lin}}(x, \theta)$ for $f(x, \theta)$ is defined as follows:

$$f_{\text{lin}}(x, \theta) := f(x, \mu) + J(x, \mu)(\theta - \mu), \tag{3}$$

where $\theta \in R^P$ and $J(x, \mu) = [\frac{\partial f(x, \theta)}{\partial \theta}]_{\theta=\mu} \in \mathbb{R}^{Q \times P}$ denotes the Jacobin matrix obtained by differentiating the function value $f(x, \theta)$ with respect to the mean parameter $\mu$. Then, one can easily see that the linearized BNN $f_{\text{lin}}(x, \theta)$ follows the Gaussian distribution, defined as follows:

$$f_{\text{lin}}(x) \sim \mathcal{N}(\boldsymbol{\mu}(x), \boldsymbol{\Sigma}(x)), \quad \boldsymbol{\mu}(x) := f(x, \mu), \quad \boldsymbol{\Sigma}(x) := J(x, \mu)\text{diag}(\sigma^2)J(x, \mu)^\top. \tag{4}$$

Based on the linearization, the KL divergence $\text{KL}\big(q(f(X_{\text{ctx}}))\|p(f(X_{\text{ctx}}))\big)$ in Eq. (2) boils down to the KL divergence between multivariate Gaussian, which has a closed-form expression.

## 3 LIMITATIONS OF THE EXISTING WORKS ON FUNCTION-SPACE BNNS

In this section, we highlight the limitations of the existing works on function-space BNNs in three perspectives: (1) the choice of priors, (2) computational complexity, and (3) the choice of context sets for KL divergence computation.

### 3.1 THE CHOICE OF PRIORS

**Gaussian process prior.** Gaussian process (GP) (Rasmussen, 2004) is a stochastic process (SP) assuming that any finite random variables of the SP follow the multivariate Gaussian distribution. The GP has been recognized as a representative function-space prior for BNNs (Sun et al., 2019; Karaletsos & Bui, 2020; Tran et al., 2022). However, using the GP prior is computationally expensive for the large and high-dimensional dataset (Liu et al., 2020a) due to the computational cost of finding the kernel hyperparameter. Thus, GP priors have been mainly used for regression tasks. It has rarely been explored for the BNNs using the commonly-used DNN architecture such as ResNet.

**Function-space prior via linearized neural network.** The linearized neural network (NN) yields a tractable function-space prior by specifying the weight-space prior $p(\theta)$ and then push-forwarding the weight-space prior to the output of the linearized NN (Rudner et al., 2022), as described in Eq. (4). However, this construction still raises concerns about using the obtained prior as the function-space regularizer $\text{KL}(q(f)\|p(f))$ because it is unclear how the mean $\boldsymbol{\mu}(x)$ and variance function $\boldsymbol{\Sigma}(x)$ would behave depending on input $x$. For instance, the mean and variance of the function-space prior corresponding to a zero-mean Gaussian weight prior $p(\theta) = \mathcal{N}(\theta; \mathbf{0}_P, \sigma^2 I_{P \times P})$ is derived as,

$$\boldsymbol{\mu}(x) = f(x, \mathbf{0}_P) = \mathbf{0}_Q, \quad \boldsymbol{\Sigma}(x) = J(x, \mathbf{0}_P)^\top \sigma^2 I_{P \times P} J(x, \mathbf{0}_P) = \sigma^2 J(x, \mathbf{0}_P)^\top J(x, \mathbf{0}_P). \tag{5}$$

However, one cannot easily interpret the behaviors of these functions. For instance, it is not clear how the variance $\boldsymbol{\Sigma}(x)$ changes according to the proximity of an input $x$ to a training set. As shown in Figs. 1a to 1c, which plots the mean and variance functions for a toy dataset, the variance remains unchanged when transitioning from IND to OOD regions.

### 3.2 COMPUTATIONAL COMPLEXITY OF LINEARIZED FUNCTION-SPACE BNNS

The tractable function-space VI using the linearized BNNs requires computing the Jacobian matrix $J(x, \mu) = [\frac{\partial f(x, \theta)}{\partial \theta}]_{\theta=\mu} \in \mathbb{R}^{Q \times P}$ every iteration to compute $\boldsymbol{\Sigma}(x)$ for $\text{KL}(q(f)\|p(f))$. Computing

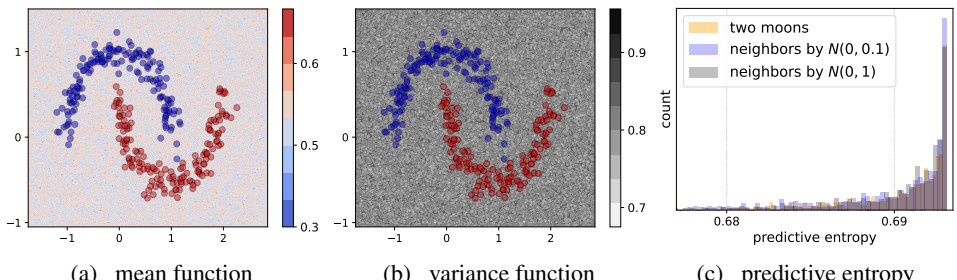

(a) mean function       (b) variance function       (c) predictive entropy

Figure 1: Using two moon classification dataset, we depict the function-space prior of the linearized BNN using residual DNN. For the weight-space prior $\mathcal{N}(\theta; \mu, \Sigma)$ with $\mu \approx 0_P$ and $\Sigma = 10 I_{P \times P}$, **Panel (a)** shows $\mathrm{E}[\mathrm{softmax}(f_j(x))]$ for sample functions $\{f_j(x)\}_{j=1}^{100} \sim \mathcal{N}(\boldsymbol{\mu}(x), \boldsymbol{\Sigma}(x))$ of Eq. (4). **Panel (b)** shows the corresponding $\mathrm{Var}[\mathrm{softmax}(f_j(x))]$. **Panel (c)** compares the predictive entropy $\mathrm{H}(p(y|x))$ on training inputs ($\bullet, \bullet$) and their neighbors obtained by adding noise $\epsilon \sim \mathcal{N}(0, 0.1)$ and $\mathcal{N}(0, 1)$. Although the function-space prior can impose varying levels of uncertainty for each input $x$, the resulting uncertainties fail to distinguish between training data and out-of-distribution data.

$J(x, \mu)$ requires GPU memory $\mathcal{O}(BPQ)$ where $B$ is a batch size, $P$ is the number of parameters, and $Q$ is the number of function outputs, which amounts to storing the gradients from $BQ$ models at each iteration during training. Thus, training function-space BNNs for a DNN with large $P$ requires prohibitively large GPU memory, which can easily lead to out-of-memory issue (as detailed in Appendix A.1). A practical solution is to treat only a subset of the parameters as random variables, such as restricting randomness to the last layer while keeping the parameters of the earlier layers deterministic. While this approach alleviates memory complexity, it might reduce flexibility in the resulting BNN model.

### 3.3 The choice of context sets

As reviewed in Section 2, evaluating the KL divergence between stochastic processes necessitates the use of the context set $X_{\mathrm{ctx}}$. The prior works show that well-curated context sets resembling the original training data yet not identical can enhance the model's uncertainty estimation capabilities (Antorán et al., 2023; Lopez et al., 2023). However, previous works underscore that the arbitrarily chosen context set without considering its relationship with the training set may have detrimental effects on model training. Indeed, we investigate how varying context set $X_{\mathrm{ctx}} = (1-\alpha)X_{\mathrm{train}} + \alpha X_{\mathrm{ext}}$ for $\alpha \in (0, 1]$ affects the performance of the function-space VI in Appendix B.1.2 and observe that its performance on IND set tends to degrade as the context set is set as external set $X_{\mathrm{ext}}$.

## 4 An Adaptive Function-Space Priors from Learning Trajectories

In this section, we introduce a novel function-space prior designed to address the limitations discussed earlier. Specifically, we present an explicit form of function-space prior to be widely used for DNN architectures. We consider DNNs as Bayesian Last-layer models, yielding the closed form of the function-space prior, and then devise the explicit mean and variance function of prior to adaptively produce higher uncertainty as GP prior does. The parameters of mean and variance functions are set leveraging the weight and feature statistics obtained from the leaning trajectory. Additionally, based on our variance function, we propose a straightforward way to compute the context feature eliminating the need for external datasets as required in previous approaches. Fig. 2a describes the procedure of prior construction and Figs. 2b and 2c describes the effect of the designed function-space prior, which is distinct from the push-forwarded prior in Figs. 1a and 1b.

Let us first decompose a neural network $f(x, \theta)$ as $f(x, \theta) = \Theta^{(L)} h(x)$ where $h(x) \in R^H$ is a *deterministic* feature extractor and $\Theta^{(L)} \in R^{Q \times H}$ is a *random* weight matrix for the linear layer. We denote $\theta^{(L)} := \mathrm{vec}(\Theta^{(L)})$ to be the vectorized weight matrix. Then, we first collect statistics required for $h(x)$ and $\theta^{(L)}$ from a learning trajectory following the procedure that will be described

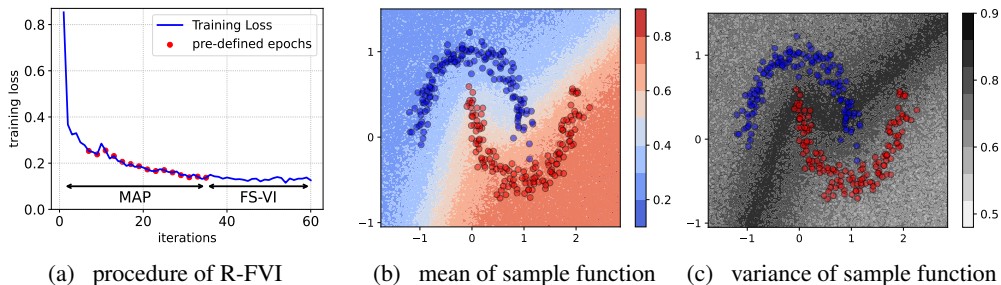

(a)  procedure of R-FVI    (b)  mean of sample function    (c)  variance of sample function

Figure 2: Using the same setting described in Fig. 1, we explore the designed function-space prior. **Panel (a)** depicts the procedure of R-FVI, where the feature $h(x)$ and weight parameters $\theta^{(L)}$ are collected at pre-defined epochs (●). **Panel (b)** depicts $\mathrm{E}[\mathrm{softmax}(f_j(x))]$ for 100 sample functions $\{f_j(x)\}_{j=1}^{100} \sim p(f(x); \boldsymbol{\mu}(x), \boldsymbol{\Sigma}(x))$ in Eq. (6). **Panel (c)** depicts the corresponding $\mathrm{Var}[\mathrm{softmax}(f_j(x))]$. Notably, our function-space prior induces the equal predictive mean in **Panel (b)** and higher variance in **Panel (c)** as the inputs are closely located on decision boundary.

briefly, and define the function-space prior as Gaussian, $f(x) \sim \mathcal{N}(\boldsymbol{\mu}(x), \boldsymbol{\Sigma}(x))$, where

$$\boldsymbol{\mu}(x) = (\widehat{\mu}_k^\top \widehat{h}(x))_{k=1}^Q, \quad \boldsymbol{\Sigma}(x) = \mathrm{diag}\left(\left(2\|m_{q_x}\|_{\widehat{\sigma}_k^2}^2 - \|\widehat{h}(x)\|_{\widehat{\sigma}_k^2}^2\right)_{k=1}^Q\right). \quad (6)$$

Here, $\widehat{h}(x)$ is the feature extractor constructed from the feature statistics of $h(x)$ evaluated from different checkpoints in a learning trajectory and $(\widehat{\mu}_k, \widehat{\sigma}_k)_{k=1}^Q$ are the class-wise weight-space statistics of $\theta^{(L)}$ computed from the same checkpoints. Below, we describe how $\widehat{h}(x)$ and $(\widehat{\mu}_k, \widehat{\sigma}_k)_{k=1}^Q$ are specified and explain the rationale behind our design choices for prior $\mathcal{N}(\boldsymbol{\mu}(x), \boldsymbol{\Sigma}(x))$.

We divide our training procedure into two phases: **phase I** where we run a vanilla SGD and collect statistics from the checkpoints on the SGD trajectory, yielding the proposed function-space prior, and **phase II** where we apply function-space VI based on the prior constructed in the first phase.

### 4.1   PHASE I: PRIOR CONSTRUCTION.

**Computing feature and weight statistics.**   To compute the statistics required for our prior, we apply the Stochastic Weight Averaging Gaussian (SWAG) (Maddox et al., 2019) which constructs an approximate Gaussian posterior $p(\theta|\mathcal{D}) \approx \mathcal{N}(\theta; \mu_{\mathrm{swag}}, \Sigma_{\mathrm{swag}})$ where $\mu_{\mathrm{swag}} := \frac{1}{T}\sum_{t=1}^T \theta(t)$ and $\Sigma_{\mathrm{swag}} := \frac{1}{T}\sum_{t=1}^T (\theta(t) - \mu_{\mathrm{swag}})(\theta(t) - \mu_{\mathrm{swag}})^\top$ for a set of checkpoints $\{\theta(t)\}_{t=1}^T$ (periodically) collected from a SGD trajectory.

Employing this idea in **phase I** of our prior construction, we first run a vanilla SGD and collect the checkpoints for a pre-defined set of epochs $\mathcal{T} := \{t_1, \ldots, t_{\mathrm{pre}}\}$. For each $t \in \mathcal{T}$, we then compute the class-wise mean features $m_k(t)$ for $k \in \{1, \ldots, Q\}$ and the diagonal total covariance $s(t)$,

$$m_k(t) = \frac{1}{N_k}\sum_{i:y_i=k} h(x_i), \quad s(t) = \frac{1}{N}\sum_{k=1}^Q \sum_{i:y_i=k} [\Delta_k(x_i)]^{\otimes 2}, \quad \Delta_k(x) = h(x) - m_k, \quad (7)$$

Here, $h(\cdot)$ is the feature extractor using the checkpoint $\theta(t)$ and $N_k := |\{i|y_i = k\}|$[2]. The $\otimes 2$ denotes the element-wise square. Along with the class-wise feature means and total variance, we also store the last-layer weight parameter $\theta^{(L)}(t)$ for later use.

After the $t_{\mathrm{pre}}$ epochs of SGD training, we compute the following time-averages of class-wise means $\{m_k\}_{k=1}^Q$ and the corresponding total covariance matrix $\mathrm{diag}(s)$,

$$m_k = \frac{1}{|\mathcal{T}|}\sum_{t\in\mathcal{T}} m_k(t), \qquad s = \frac{1}{|\mathcal{T}|}\sum_{t\in\mathcal{T}} s(t). \quad (8)$$

---

[2]For regression task, since $m_k(t)$ and $S(t)$ can not be directly defined due to the real-valued label, we use a newly defined pseudo label by discretizing the real-valued space into $Q$ intervals, as described in Appendix A.5.

Similarly, for the last-layer parameters $\theta^{(L)}(t)$, we compute the time-averages of empirical mean $\widehat{\mu}$ and diagonal covariance $\text{diag}(\widehat{\sigma}^2)$.

$$\widehat{\mu} = \frac{1}{|\mathcal{T}|}\sum_{t \in \mathcal{T}}\theta^{(L)}(t), \qquad \widehat{\sigma}^2 = \frac{1}{|\mathcal{T}|}\sum_{t \in \mathcal{T}}\theta^{(L)}(t)^{\otimes 2} - \widehat{\mu}^{\otimes 2} \tag{9}$$

**Constructing feature extractor.** Given the statistics, the feature extractor $\widehat{h}(x)$ is defined as a mixture model over $Q$ classes,

$$\widehat{h}(x) = \sum_{k=1}^{Q} w_k(x)\, m_k \in \mathbb{R}^H, \qquad w_k(x) = \frac{\exp(-\|\Delta_k(x)\|_{s^{-1}}^2)}{\sum_{j=1}^{Q}\exp(-\|\Delta_j(x)\|_{s^{-1}}^2)}. \tag{10}$$

where $\|\Delta_k(x)\|_{s^{-1}}^2 = \Delta_k(x)^\top \text{diag}(s)^{-1}\Delta_k(x)$ denotes Mahalanobis distance (MHD) using $\Delta_k(x)$ in Eq. (7) and $\{w_k(x)\}_{k=1}^{Q} \in [0,1]$ denotes the weight vectors satisfying $\sum_{k=1}^{Q} w_k(x) = 1$.

**Constructing function-space prior.** Now we describe our function space prior Eq. (6) computed from the feature extractor $\widehat{h}(x)$ and weight statistics $(\widehat{\mu}_k, \widehat{\sigma}_k)$, and explain the motivation behind the prior's construction and the properties obtained.

For the mean function $\boldsymbol{\mu}(x)$, we simply take it to be an inner-product between the feature extractor and checkpoint mean of the linear layer,

$$\boldsymbol{\mu}(x) = \left(\widehat{\mu}_k^\top \widehat{h}(x)\right)_{k=1}^{Q},$$

where $\widehat{\mu}_k$ denotes the elements of $\widehat{\mu}$ corresponding to the $k^{\text{th}}$ class, i.e., the expected $k$-th row $\text{E}[\Theta^{(L)}]_{k,:} = \widehat{\mu}_k$ for matrix form $\Theta^{(L)} \in R^{Q \times H}$. Note that this is equivalent to the mean of the linearized function space BNN $f(x) = \Theta^{(L)}\widehat{h}(x)$, i.e, $\text{E}[\Theta^{(L)}\widehat{h}(x)]_k = \widehat{\mu}_k^\top \widehat{h}(x)$ for $k = 1, .., Q$.

For the covariance function $\boldsymbol{\Sigma}(x)$, we consider

$$\boldsymbol{\Sigma}(x) = \text{diag}\left(\left(2\|m_{q_x}\|_{\widehat{\sigma}_k^2}^2 - \|\widehat{h}(x)\|_{\widehat{\sigma}_k^2}^2\right)_{k=1}^{Q}\right) \quad \text{with} \quad q_x := \underset{k \in \{1,...,Q\}}{\arg\max}\, w_k(x),$$

though this may seem non-trivial. Intuitively, given $\widehat{h}(x)$, this finds the nearest feature $m_{q_x}$ over $\{m_k\}_{k=1}^{Q}$. Then, this computes the gap between function-space variances of $\widehat{h}(x)$ and $m_{q_x}$ using $f(x) = \Theta^{(L)}\widehat{h}(x)$, i.e., $\text{Var}[\Theta^{(L)}m_{q_x}]_k = \|m_{q_x}\|_{\widehat{\sigma}_k^2}^2$ and $\text{Var}[\Theta^{(L)}\widehat{h}(x)]_k = \|\widehat{h}(x)\|_{\widehat{\sigma}_k^2}^2$. Through this form, we intend the $\boldsymbol{\Sigma}(x)$ to produce higher variance as $\widehat{h}(x)$ is less close to its vicinity $m_{q_x}$. Also, we observe that $\boldsymbol{\Sigma}(x)$ shares a similar structure with the predictive variance of Gaussian processes (Rasmussen, 2004),

$$\boldsymbol{\Sigma}_{\text{GP}}(x) = k(x,x) - k(x,X)K(X,X)^{-1}k(X,x),$$

where the first term $k(x,x)$ roughly matches with $2\|m_{q_x}\|_{\widehat{\sigma}_k}^2$ in derived from our choice as prior. The second term $k(x,X)K(X,X)^{-1}k(X,x)$ has a similar role to the term $\|\widehat{h}(x)\|_{\widehat{\sigma}_k^2}^2$ in the sense that the variance on $x$ can be modeled by training inputs $X$ and mixture features $\{m_k\}_{k=1}^{Q}$. Below, we describe the property of our prior that $\boldsymbol{\Sigma}(x)$ produces higher variance as an feature $\widehat{h}(x)$ deviates from its vicinity mixture component $m_{q_x}$.

**Proposition 4.1.** *(informal) For two input $x_1, x_2 \in \mathcal{X}$ and features $\widehat{h}(x_1), \widehat{h}(x_2) \in R^H$, let $k = q_{x_1} = q_{x_2}$ for some $k = \{1, .., Q\}$ meaning $m_k$ is their vicinity feature. Then, if $\widehat{h}(x_1)$ is not equal to but closer to $m_k$ than $\widehat{h}(x_2)$ in terms of MHD, i.e, $a_k < w_{q_{x_2}} < w_{q_{x_1}} < 1$ for $a_k < 1$ (specified in Appendix), each $i$-th variance of $\boldsymbol{\Sigma}(x_1)$ is larger than that of $\boldsymbol{\Sigma}(m_k)$ and smaller than that of $\boldsymbol{\Sigma}(x_2)$,*

$$[\boldsymbol{\Sigma}(m_k)]_i < [\boldsymbol{\Sigma}(x_1)]_i < [\boldsymbol{\Sigma}(x_2)]_i \quad \text{for} \quad i = 1, .., Q, \tag{11}$$

*intuitively meaning if $m_k$ is likely to be in-distribution feature, then $\boldsymbol{\Sigma}(x_2)$ would have higher variance because $\widehat{h}(x_2)$ is farther away from $m_k$.*

*Proof.* Concrete statement with assumption and its proof can be checked in Appendix A.4 □

---

**Algorithm 1** Function-Space VI using the prior of Eq. (6) and adversarial feature of Eq. (13)

---

**Require:** Pre-defined epoch $\mathcal{T}$, extractor parameter $\theta$, last-layer variational parameter $(\mu, \sigma)$

---

1: **for** $t = 1, \dots, T$ **do**
2:     **if** $t \le t_{\text{pre}}$   **// PHASE I: PRIOR CONSTRUCTION**
3:         Set last-layer parameter $\theta^{(L)} = \mu^{(L)}$, and Train $\theta$ and $\theta^{(L)}$ by $\mathcal{L}_{\text{fvi}}$ of Eq. (1) without KL;
4:         **if** $t \in \mathcal{T}$ **then** Update $(m_k, s, \widehat{\mu}, \widehat{\sigma}_k^2)$ in Eqs. (8) and (9) recursively
5:         **if** $t = t_{\text{pre}}$ **then** Construct function-space prior $\mathcal{N}(\boldsymbol{\mu}(x), \boldsymbol{\Sigma}(x))$ by Eq. (6)
6:     **else**       **// PHASE II: FUNCTION-SPACE VI**
7:         **if** $t = t_{\text{pre+1}}$ **then** Set variational weight parameter of $L$-th layer as $\mathcal{N}(\psi^{(L)}; \mu, \sigma^2)$
8:         Sample $f_{(j)}(x_i) \sim q(f(x_i))$ in Eq. (12) in function space,
9:         Construct $\mathcal{N}(\boldsymbol{\mu}(x), \boldsymbol{\Sigma}(x))$ in Eq. (6) using $z_{\text{adv}}$, defined in Eq. (13) for $(x_i, y_i) \in \mathcal{D}$
10:        Train $\theta, \mu^{(L)}$, and $\sigma^{(L)}$ with $\mathcal{L}_{\text{fvi}}$ of Eq. (1) with KL of Eq. (2)
11: **end for**

---

## 4.2   PHASE II: FUNCTION-SPACE VI

**Function-space variation inference with the designed prior.**   Once the function-space prior is prepared, we employ function-space variational inference for training the variational parameters. We consider the function-space variational distribution $\mathcal{N}(\boldsymbol{\mu}(x), \boldsymbol{\Sigma}(x))$,

$$\boldsymbol{\mu}(x) = \left( \mu_k^\top h(x) \right)_{k=1}^Q, \qquad \boldsymbol{\Sigma}(x) = \text{diag} \left( \| h(x) \|_{\sigma_k^2}^2 \right)_{k=1}^Q \tag{12}$$

by employing the closed form of function-space distribution $f(x) = \Psi^{(L)} h(x)$ with the feature extractor $h(x)$, variational last-layer random weight $\psi^{(L)} \sim \mathcal{N}(\mu, \sigma^2)$, and its matrix form $\Psi^{(L)}$, where feature extractor parameter $\theta$ and variational parameters $(\mu, \sigma)$ are trained. Similarly to the function-space prior, the $\mu_k$ and $\sigma_k$ denote the partial elements of $\mu$ and $\sigma$ for $k^{\text{th}}$ class, respectively.

**Adversarial context feature.**   Additionally, we propose the adversarial context feature to compute the function-space KL-divergence in Eq. (2) without relying on external dataset for the context set $X_{\text{ctx}}$. As the proposed function-space prior is designed to induce larger variance when $\widehat{h}(x)$ is farther from the closest feature $m_{q_x}$ meaning that the corresponding $w_{q_x}$ decreases. Based on this intuition, we seek the context feature that are adversarially minimizing $w_{q_x}(x)$. Unlike the typical adversarial attacks where the search is done at the input space, we do this at the feature level. Specifically, let $w_{q_x} := w'_{q_x} \circ h$, and we define an adversarial hidden feature $z_{\text{adv}} := \arg\min_{z \in B_r(h)} w'_{q_x}(z)$ and computed it approximately as

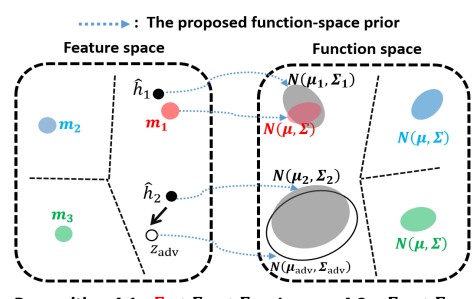

: The proposed function-space prior

**Proposition 4.1 :** $\boldsymbol{\Sigma} < \boldsymbol{\Sigma_1} < \boldsymbol{\Sigma_2}$,   **Lemma 4.2 :** $\boldsymbol{\Sigma_2} < \boldsymbol{\Sigma}_{\text{adv}}$

Figure 3: Our prior has larger variances ($\boldsymbol{\Sigma} < \boldsymbol{\Sigma_1} < \boldsymbol{\Sigma_2}$) if $\widehat{h}_1$ is closer to $m_1$ than $\widehat{h}_2$ in sense of MHD. Our feature $z_{\text{adv}}$ is located to induce larger variance ($\boldsymbol{\Sigma_2} < \boldsymbol{\Sigma}_{\text{adv}}$).

$$z_{\text{adv}} \approx h - r \, \text{sign} \left( \nabla_h \log w'_{q_x}(h) \right) \in R^H, \tag{13}$$

using Fast Gradient Sign Attack (FSGM) (Goodfellow et al., 2014). The obtained feature $z_{\text{adv}}$ can be used instead of the original feature $\widehat{h}(x)$ in Eq. (6) in computing the function-space KL divergence during variational inference. We state the property of $z_{\text{adv}}$ in Lemma 4.2.

**Lemma 4.2.**  *For input $x \in \mathcal{X}$ and its smoothed hidden feature $\widehat{h}(x) \in R^H$, the adversarial hidden feature $z_{adv}$ is located to increase the variance of the prior, i.e., $[\boldsymbol{\Sigma}(x)]_i < [\boldsymbol{\Sigma}_{adv}]_i$ for all $i$, where $\boldsymbol{\Sigma}_{adv}$ denotes the variance of function-space prior obtained by replacing $\widehat{h}(x)$ with $z_{adv}$ in Eq. (6).*

We refer to the proposed method as the Refined function-space VI (R-FVI) using Learning Trajectory-based function-space prior. To aid the understanding, we illustrate the effect of the proposed prior and context feature in Fig. 3, and describe the training procedure of R-FVI in Algorithm 1.

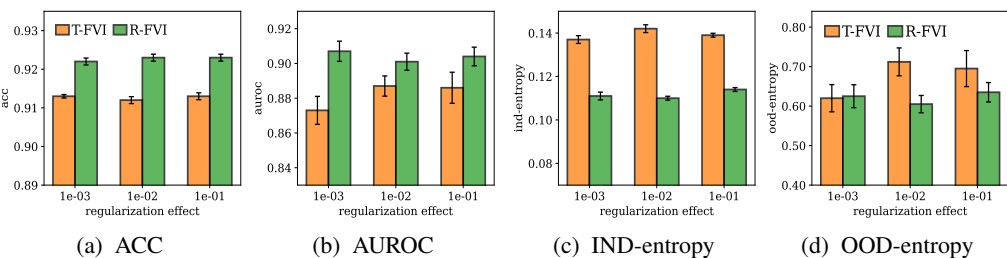

Figure 4: Figs. 4a to 4d compares ACC, AUROC, Predictive entropy on IND set (CIFAR 10) and OOD set (SVHN) over varying KL regularization hyperparameter $\lambda$. The R-FVI obtains the higher ACC and AUROC for all $\lambda$ by yielding smaller predictive entropy on IND sets.

## 5 RELATED WORK

**Function-space BNN, VI, and Prior.**    Our work aligns with prior works (Sun et al., 2019; Rudner et al., 2022; 2023; Lin et al., 2023) presenting the function-space VI. Our work relieves their limitations by presenting the explicit form of function-space prior using learning trajectory. Additionally, our approach aligns with prior works (Hafner et al., 2020; Flam-Shepherd et al., 2017; Tran et al., 2022) in designing function-space priors. Unlike Hafner et al. (2020), which uses noise perturbation input for prior construction, we consider adversarial perturbation in feature space and clarify its impact on the function-space prior. Furthermore, unlike Flam-Shepherd et al. (2017); Tran et al. (2022), which use GP priors primarily for shallow BNNs, our function-space prior is designed to be feasible with large-scale BNNs using ResNet He et al. (2016) and VIT Dosovitskiy et al. (2020). Unlike Liu et al. (2020b) directly using the approximate GP prior into DNN's last-layer, our work designs the covariance function motivated from GP predictive variance.

**Empirical Bayes for BNNs.**    Empirical Bayes estimates the parameters of the prior distribution through training. This contrasts with the conventional Bayesian approach where prior parameters are set in advance Casella (1992). For BNNs, Immer et al. (2021) employs marginal likelihood optimization for training the prior. Krishnan et al. (2020) uses the parameters of the pre-trained model as the mean parameters of the weight-space prior. Shwartz-Ziv et al. (2022) uses re-scaled parameters of pre-trained models as the weight-space prior for transfer learning. However, unlike theses work, our work uses the parameter trajectory during training to construct the function-space prior. Furthermore, our prior is developed from scratch training, whereas Krishnan et al. (2020); Shwartz-Ziv et al. (2022) relies on pre-trained parameters on training or upstream datasets as prior.

**Implicit Process.**    Our work shares similarities with variants of the variational implicit process (VIP) Ma et al. (2019); Ma & Hernández-Lobato (2021); Rodrguez-Santana et al. (2022); Ortega et al. (2022) in modeling stochastic functions using DNNs. However, while VIP variants aim to enhance modeling capabilities by constructing implicit distributions with stochastic NN generators Ma & Hernández-Lobato (2021) and sparse GPs Rodrguez-Santana et al. (2022), our focus is on building effective function-space prior to improve BNNs.

## 6 EXPERIMENTS

**Experiment Setting.**    We basically use widely-adopted DNN architectures, such as ResNet (He et al., 2016), as our base model. Then, we convert the model into a last-layer BNN by replacing the last MLP layer with a Bayesian MLP layer due to memory constraints as described in Section 3.

To evaluate the trained model, we measure the test accuracy (ACC), negative log likelihood (NLL), and expected calibration error (ECE) on the IND test set as indicators of uncertainty estimation performance for the IND set. Also, we measure the Area Under the Receiver Operating Characteristic (AUROC) on the OOD set, serving as indicators of performance for OOD set. We use the predictive entropy as the input and the IND set's status as the label.

### 6.1 FUNCTION-SPACE PRIOR INDUCING VARYING LEVEL OF UNCERTAINTY

**Uncertainty of the function-space prior.**    We investigate whether the proposed function-space prior induces varying levels of uncertainty depending on each function's input. We train the ResNet

Table 1: We report each metric using Bayesian model Averaging with 10 sample functions ($J{=}10$) and 3 random seeds; **boldface** and underline denote the first and second-best metrics, respectively. For T-FVI, we use CIFAR-100 and Tiny-ImageNet as the context set, respectively.

| Model / Data | Method | ACC ↑ | NLL ↓ | ECE ↓ | AUROC ↑ |
|---|---|---|---|---|---|
| ResNet 18 CIFAR 10 | MAP | (0.948, 0.003) | (0.199, 0.011) | (0.029, 0.000) | (0.939, 0.007) |
| | SWAG | (0.942, 0.002) | (0.195, 0.008) | (**0.024**, 0.001) | (0.914, 0.002) |
| | SNGP | (0.914, 0.002) | (0.407, 0.008) | (0.060, 0.001) | (**0.993**, 0.001) |
| | WVI (FL) | (0.909, 0.001) | — | (0.048, 0.003) | (0.918, 0.009) |
| | WVI (LL) | (0.950, 0.002) | (0.216, 0.001) | (0.030, 0.003) | (0.922, 0.014) |
| | T-FVI | (0.947, 0.002) | (0.207, 0.011) | (0.032, 0.002) | (0.938, 0.012) |
| | R-FVI (**our**) | (**0.952**, 0.001) | (**0.187**, 0.005) | (0.028, 0.001) | (0.956, 0.004) |
| ResNet 50 CIFAR 100 | MAP | (0.797, 0.001) | (0.835, 0.002) | (0.074, 0.002) | (0.805, 0.014) |
| | SWAG | (0.772, 0.002) | (0.918, 0.008) | (0.077, 0.003) | (**0.896**, 0.001) |
| | WVI (LL) | (0.780, 0.004) | (1.148, 0.012) | (0.099, 0.002) | (0.777, 0.028) |
| | T-FVI | (0.794, 0.001) | (0.846, 0.006) | (0.076, 0.002) | (0.846, 0.015) |
| | R-FVI (**our**) | (**0.799**, 0.003) | (**0.792**, 0.012) | (**0.056**, 0.002) | (0.850, 0.015) |

20 using R-FVI on CIFAR 10 and obtain the prior with pre-defined epoch $\mathcal{T} = \{0.8T - 20, 0.8T - 16, 0.8T - 12, 0.8T - 8, 0.8T - 4\}$ with $T = 200$

Fig. 5a shows the averaged $w_{q_x}$ of Eq. (10), representing the distance between $\widehat{h}(x)$ and its closest feature $m_{q_x}$, for the IND set (CIFAR 10), OOD set (SVHN), and the adversarial feature $z_{\text{adv}}$ of Eq. (13) with radius $r \in \{.05, .10, .20\}$. Fig. 5b shows the corresponding averaged standard deviation of the function-space prior, i.e, $\text{Tr}(\mathbf{\Sigma}^{\frac{1}{2}}(x))$ in Eq. (6). These figures demonstrate that the proposed prior induces higher uncertainty in model's output when $w_{q_x}$ decreases, which is stated in Proposition 4.1 and Lemma 4.2. We also investigate other priors derived in different SGD trajectories in Appendix B.1.1, confirming that the prior of each trajectory exhibits a similar trend when pre-trained epoch is set after $0.5T$ epoch. Fig. 5c shows that the obtained function-space prior produces more uncertain predictive sample functions when the inputs are OOD data point.

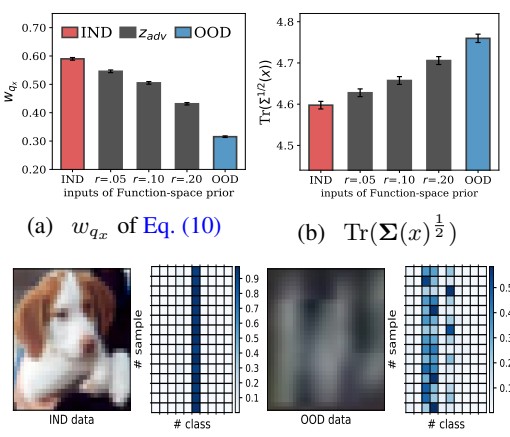

(a) $w_{q_x}$ of Eq. (10)   (b) $\text{Tr}(\mathbf{\Sigma}(x)^{\frac{1}{2}})$

(c) random predictions sampled from our prior

Figure 5: Investigation on the proposed prior

**Investigation on the effect of the KL regularization.** We investigate the effect of the function-space KL regularization on the performance on IND and OOD sets. For comparison, we consider T-FVI with the uniform function-space prior $\mathcal{N}(0, 10I_{Q \times Q})$; 10 is empirically found over $\{5, 10, 50\}$ and set the context set as CIFAR 100 as following the experiment setting in Rudner et al. (2022). We consider the KL regularization hyperparameter $\lambda \in \{10^{-3}, 10^{-2}, 10^{-1}\}$ in Eq. (2) as the relative ratio between likelihood and KL term to apply the same amount of the regularization into the model during training, regardless of scale of KL term; $\lambda = 10^{-1}$ means the value of KL term is adaptively rescaled to be $1/10$ of the likelihood over iterations.

Fig. 4 compares the ACC, AUROC, Predictive entropy on IND set (CIFAR 10) and OOD set (SVHN) over different $\lambda$. These results imply the KL regularization of R-FVI via our function-space prior leads to better accuracy and AUROC for all $\lambda$, as shown in Figs. 4a and 4b. Notably, the KL regularization of R-FVI allows its variational distribution to have smaller predictive entropy on the IND and OOD set as shown in Figs. 4c and 4d while yielding the better OOD performances for all $\lambda$.

### 6.2 IMAGE CLASSIFICATION TASK

Following the experimental setup conducted in (Rudner et al., 2022), we perform the classification tasks using ResNet 18 and 50 to demonstrate the effectiveness of R-FVI. We compare the proposed inference with other baseline inference methods. Further details can be found in Appendix B.2.

Table 2: We report the mean and one-standard deviation of each metric over 3 random seeds. We set the context set as Tiny-ImageNet for training the down-stream torch vision datasets.

| Dataset | Method | ACC ↑ | NLL ↓ | ECE ↓ | AUROC-S ↑ | AUROC-C ↑ |
|---|---|---|---|---|---|---|
| PETS 37 | MAP | (0.940, 0.002) | (0.279, 0.005) | (0.038, 0.001) | (1.000, 0.000) | (0.998, 0.000) |
| | T-FVI | (0.937, 0.001) | (0.225, 0.001) | (0.015, 0.002) | (1.000, 0.000) | (**0.999**, 0.000) |
| | R-FVI | (**0.942**, 0.001) | (**0.215**, 0.003) | (**0.010**, 0.001) | (1.000, 0.000) | (**0.999**, 0.000) |
| DTD 47 | MAP | (0.790, 0.006) | (1.068, 0.016) | (0.131, 0.004) | (0.972, 0.024) | (0.965, 0.006) |
| | T-FVI | (0.785, 0.009) | (0.801, 0.022) | (**0.029**, 0.004) | (**0.988**, 0.002) | (**0.985**, 0.004) |
| | R-FVI | (**0.793**, 0.001) | (0.795, 0.022) | (0.033, 0.004) | (0.988, 0.006) | (0.983, 0.001) |
| AIRCRAFT 100 | MAP | (0.701, 0.005) | (1.157, 0.008) | (0.094, 0.002) | (0.998, 0.001) | (0.995, 0.001) |
| | T-FVI | (0.711, 0.000) | (1.155, 0.000) | (0.124, 0.000) | (**0.999**, 0.000) | (**0.998**, 0.000) |
| | R-FVI | (**0.718**, 0.006) | (**1.055**, 0.033) | (**0.045**, 0.008) | (**0.999**, 0.000) | (**0.998**, 0.001) |

**Results.** Table 1 demonstrates that R-FVI generally outperforms the baselines in terms of ACC, NLL, and ECE on the IND set. Especially, R-FVI is more effective when using ResNet 50, i.e., the larger model. For OOD performance, R-FVI outperforms other baselines except SGNP (Liu et al., 2020b) using the approximate GP prior in the last-layer. Additionally, we confirm the variance property of our priors in Appendix B.2.1 and investigate how the performance of R-FVI may vary depending on the trajectory $\mathcal{T}$ and radius $r$ of $z_{\text{adv}}$ in Appendix B.2.2. The SGNP trained on CIFAR-100 cannot compared directly because the trained SGNP appear to be significantly underfitted, even after testing various kernel hyperparameters as shown Appendix B.2.3.

### 6.3 TRANSFER LEARNING WITH VISION TRANSFORMER.

We demonstrate the effectiveness of R-FVI for transfer learning using a large-scale pre-trained model. We use the pre-trained VIT-Base model Dosovitskiy et al. (2020), using 16 patch and 224 resolution, trained on ImageNet 21K [3]. We consider the last-layer BNN as done in ResNet.

**Results.** Table 2 demonstrates that R-FVI results in reliable uncertainty estimation on each IND set and OOD sets (SVHN and CIFAR 100) when adapting the large-sized VIT model (#parameters = $86.6M$) to downstream task . Additional results of different trajectories are reported in Appendix B.3.

### 6.4 UCI REGRESSION TASK.

We also conduct a UCI regression task to showcase the effectiveness of R-FVI. Since the MHD cannot be used for real-valued labels, we employ a slight modification employing $K$ bins defined in function space for obtaining the discrete pseudo-label, as described in Appendix A.5.

**Results.** Fig. 6 indicates that R-FVI generally outperforms other baselines. Also, the consistency of performance across different number of bins ($K$) can be checked in Appendix B.4.

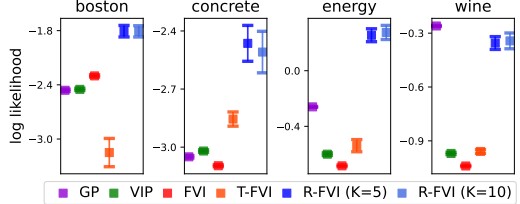

Figure 6: Log likelihood for UCI regression tasks.

### 7 CONCLUSION

We propose an explicit form of function-space prior that can be easily used with the widely-used DNN architectures, as well as to adaptively assign higher uncertainty for each function's output. We demonstrate that our prior is effective in improving uncertainty estimation, especially for the large-sized model.

However, our method has some limitations. As our prior utilizes information from pre-trained epochs, the function-space prior and its variational posterior depend on the selected pre-trained epoch. Thus, tuning the pre-trained epochs is necessary. For the regression task, our prior requires binning to obtain the pseudo-discrete labels from real-valued outputs.

---

[3]https://github.com/huggingface/pytorch-image-models

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

# A  APPENDIX: METHODOLOGY DETAILS

## A.1  COMPUTATIONAL COMPLEXITY OF T-FVI AND R-FVI

**Computational complexity of T-FVI.**  Training a function-space BNN by variational requires to compute the (1) the expected log likelihood term and (2) KL divergence in ELBO, as described in Eq. (14).

$$\mathrm{E}_{q(f)}\left[\sum_{n=1}^{N}\log p\big(y_n|f(x_n)\big)\right] - \lambda\,\mathrm{KL}\big(q(f)\|p(f)\big), \tag{14}$$

where the KL divergence $\mathrm{KL}\big(q(f)\|p(f)\big) = \sup_{X_{\mathrm{ctx}}\in\mathcal{X}^M}\mathrm{KL}\big(\,q(f(X_{\mathrm{ctx}},\phi)\,\|\,p(f(X_{\mathrm{ctx}},\theta))\,\big)$ is computed using the following approximation:

$$\mathrm{KL}\big(\,q(f(X_{\mathrm{ctx}},\phi)\,\|\,p(f(X_{\mathrm{ctx}},\theta))\,\big) \approx \mathrm{KL}\big(\mathcal{N}(\boldsymbol{\mu}_\phi(X_{\mathrm{ctx}}),\boldsymbol{\Sigma}_\phi(X_{\mathrm{ctx}}))\,\|\,\mathcal{N}(\boldsymbol{\mu}_\theta(X_{\mathrm{ctx}}),\boldsymbol{\Sigma}_\theta(X_{\mathrm{ctx}}))\big),$$

where $(\boldsymbol{\mu}_\phi(X_{\mathrm{ctx}}),\boldsymbol{\Sigma}_\phi(X_{\mathrm{ctx}}))$ are the mean and covariance of the approximate variational function-space distribution $q(f)$ obtained by using the linearization of Eq. (3) with variational weight parameter $\phi$. The $(\boldsymbol{\mu}_\theta(X_{\mathrm{ctx}}),\boldsymbol{\Sigma}_\theta(X_{\mathrm{ctx}}))$ denotes those of the corresponding function-space prior obtained by using prior weight parameter $\theta$.

The main computational bottleneck for computing the ELBO in Eq. (14) is to compute the Jacobian matrix

$$J(\cdot,\mu) = [\frac{\partial f(\cdot,\theta)}{\partial\theta}]_{\theta=\mu} \in \mathbb{R}^{Q\times P},$$

used for $\boldsymbol{\Sigma}(\cdot){=}J(\cdot,\mu)\mathrm{diag}(\mathrm{diag}(\sigma^2))J(\cdot,\mu)^\top$. This is because computing the $J(\cdot,\mu)$ requires GPU memory proportional to $\mathcal{O}(BPQ)$, where $B$ is the batch dataset size, $P$ is the number of model parameters, and $Q$ is the dimension of the function output. The amount of GPU memory can be understood as the accumulation of gradients from $BQ$ models at each iteration for Jacobian computation.

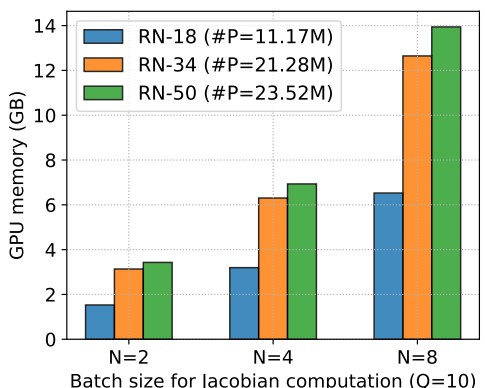

Figure 7: GPU Memory for Jacobian computation

Indeed, as computing the Jacobian matrix for the widely-used DNN architectures, such as ResNet 18, 34, and 50, these models face the issue of the Out-of-GPU memory easily. We demonstrate the amount of GPU memory used for computing the Jacobian over varying batch sizes ($N = 2, 4, 8$) and $Q = 10$ in Fig. 7. This figure potentially sheds light on the challenges associated with considering the function-space distribution of large-scale fully BNN models via Jacobian computation.

**Computational complexity of R-FVI**  To address this issue, the proposed R-FVI considers using last-layer BNNs, assuming the last layer is the only Bayesian layer. This approach can reduce the computational memory from $\mathcal{O}(BPQ)$ to $\mathcal{O}(BP_LQ)$ by computing the Jacobian of the last layer, which consists of $P_L$ parameters with $P_L \ll P$; with this reason, the tractable FVI also employs the Jacobian matrix of the last layer for the KL divergence computation, as described in Rudner et al. (2022).

Additionally, if the last layer is a Bayesian MLP layer, the Jacobian matrix can be computed analytically without using a large amount of GPU memory. Therefore, we can construct the function-space distribution for large-scale BNNs.

Furthermore, for the last-layer hidden feature $h = f^{(L-1)} \circ \cdots \circ f^{(1)}(x) \in \mathbb{R}^H$, where $H$ is the dimension, the R-FVI uses $\mathcal{O}(H(Q + 1))$ memory for the last-layer hidden feature parameters of Eq. (8) and $\mathcal{O}(2HQ)$ for the last-layer weight parameters of Eq. (9). By updating these empirical parameters in an online batch manner, R-FVI does not need to store the parameters of $|\mathcal{T}|$ trajectories during the periods of SGD iterations.

### A.2 COMPUTATION OF THE FUNCTION-SPACE DISTRIBUTION FOR THE LAST-LAYER BNNs

For an input $x \in \mathcal{X}$, we denote $f(x, \theta) \in R^Q$ as the output of the $L$-layers BNN using the random weight parameters $\theta = \{\theta^{(l)}\}_{l=1}^L$, as follows:

$$f(x, \theta) = \left( f^{(L)} \circ \cdots \circ f^{(2)} \circ f^{(1)} \right)(x), \quad \text{and} \quad f^{(l)}(x) = \sigma(\Theta^{(l)} [\, x \,;\, 1 ]), \qquad (15)$$

where $\theta^{(l)}$ denotes $l$-th layer random weight parameters including the bias parameter, and $\sigma(\cdot)$ denotes the activation function. We omit the bias term of each $\theta^{(l)}$, which does not raise the issue of our statement.

To detour the memory issue of the Jacobin computation described in Appendix A.1, we assume $f(x, \theta)$ to follow the specific structure as described in Assumption A.1.

**Assumption A.1.** The $f(x, \theta)$ is assumed to to be the last-layer BNNs following these properties:

- The first $L - 1$ layers $\{f^{(l)}\}_{l=1}^{L-1}$ are deterministic layers. In view of the random weight parameterization used in BNNs, this assumption can be understood as the $l$-th random weight parameter $\theta^{(l)}$ follows the Dirac delta distribution using parameter $\mu^{(l)}$, i.e., $p(\theta^{(l)}) = \delta_{\mu^{(l)}}(\theta^{(l)})$.

- The last $L$-th layer $f^{(L)}$ is a Bayesian MLP layer using Gaussian random weight parameter $\theta^{(L)}$, i.e., $\text{vec}(\Theta^{(L)}) \sim \mathcal{N}(\text{vec}(\mu^{(L)}), \text{diag}(\text{vec}(\Sigma^{(L)})))$.

Then, for the last-layer feature $h(x) = (f^{(L-1)} \circ \cdots \circ f^{(2)} \circ f^{(1)})(x) \in R^H$, we can re-express the $f(x, \theta) \in R^Q$ as follows:

$$f(x, \theta) = \Theta^{(L)} h(x) = [\, \Theta_{1,:}^{(L)} h(x) \,, \,.. \,, \, \Theta_{Q,:}^{(L)} h(x) \,] \in R^Q, \qquad (16)$$

where $\Theta_{k,:}^{(L)}$ denotes the $k$-th row of the last weight parameter $\Theta^{(L)} \in R^{Q \times H}$. Then, we can compute the parameters for the function-space distribution analytically, as described in Lemma A.2.

**Lemma A.2.** *Under the assumption of the last-layer BNN described in Assumption A.1, the function-space distribution $p(f(x; \theta)) = \mathcal{N}(\boldsymbol{\mu}(x), \boldsymbol{\Sigma}(x))$ has the following closed form of the parameters:*

$$\boldsymbol{\mu}(x) = (\mu_k^{(L)\top} h(x))_{k=1}^Q, \qquad \boldsymbol{\Sigma}(x) = \text{diag}\left( \left[ \, \|h(x)\|_{\sigma_1^2}^2 \,, \,.. \,, \, \|h(x)\|_{\sigma_Q^2}^2 \right] \right) \in R^{Q \times Q}, \quad (17)$$

*where $\sigma_k^2$ denotes the $k$-th row of $\Sigma^{(L)}$, i.e., $\sigma_k^2 = \Sigma_{k,:}^{(L)} \in R^H$.*

*Proof.* The result of $\boldsymbol{\mu}(x)$ is trivial because of $\mathrm{E}_{\theta^{(L)}}[\theta^{(L)} h(x)] = \mu^{(L)} h(x)$.

Next, we compute the Jacobin matrix $J(x, \mu) := [\frac{\partial f}{\partial \theta^{(L)}}]_{\theta^{(L)} = \mu^{(L)}} \in \mathbb{R}^{Q \times P}$, where $P$ denotes the number of the last-layer weight parameter, i.e., $P = Q \times H$. Then, the $k$-th row of Jacobian matrix $J(\cdot, \mu)_{k,:} \in R^P$ is computed as follows:

$$J(x, \mu)_{k,:} = \left[ \frac{\partial(\Theta_{1,:}^{(L)} h(x))}{\partial \Theta_{k,:}} \,, \,.... \,, \, \frac{\partial(\Theta_{k,:}^{(L)} h(x))}{\partial \Theta_{k,:}} \,, \,.... \,, \, \frac{\partial(\Theta_{Q,:}^{(L)} h(x))}{\partial \Theta_{k,:}} \right] \qquad (18)$$

$$= \left[ \underbrace{\mathbf{0}_H}_{\text{1-th}} \,, \,... \,, \, \underbrace{h(x)}_{k\text{-th}} \,, \,... \,, \, \underbrace{\mathbf{0}_H}_{Q\text{-th}} \right] \in R^P, \qquad (19)$$

which consists of the non-zero entries as $h(x) \in R^H$ in $k$-th block and zero entries $\mathbf{0}_H \in R^H$ in left blocks. Then, the $(q, p)$-th element of $\boldsymbol{\Sigma}(x) \in R^{Q \times Q}$ is computed as follows:

$$\boldsymbol{\Sigma}(x)_{q,p} = \left[ J(x, \mu) \text{diag}(\text{vec}(\Sigma)) J(x, \mu)^\top \right]_{q,p} \qquad (20)$$

$$= \underbrace{h(x)^\top \text{diag}(\sigma_q^2) h(x)}_{:= \|h(x)\|_{\sigma_q^2}^2} \mathbf{1}_{q=p} = \|h(x)\|_{\sigma_q^2}^2 \mathbf{1}_{q=p}. \qquad (21)$$

This yields that the covariance $\boldsymbol{\Sigma}(x)$ of the functions-space distribution has the following form:

$$\boldsymbol{\Sigma}(x) = \text{diag}\left( \left[ \, \|h(x)\|_{\sigma_1^2}^2 \,, \,.. \,, \, \|h(x)\|_{\sigma_Q^2}^2 \right] \right) \in R^{Q \times Q}. \qquad (22)$$

$\square$

### A.3 MOTIVATION OF THE FUNCTION-SPACE PRIOR CONSTRUCTION

Gaussian process (GP) has been the widely-used function-space prior Rasmussen (2004). The construction of our function-space prior is motivated from the GP predictive posterior distribution $p(f_{\mathcal{GP}}(x) \mid \mathcal{D}) = \mathcal{N}(\boldsymbol{\mu}(x_*), \boldsymbol{\Sigma}(x_*))$ for $X = \{x_i\}_{i=1}^N$ and $Y = \{y_i\}_{i=1}^N$, represented as,

$$\boldsymbol{\mu}(x_*) = \underbrace{(K(X, X)^{-1}\text{vec}(Y))^T}_{\text{weight}} \underbrace{K(X, x_*)}_{\text{kernel smoother}} \tag{23}$$

$$\boldsymbol{\Sigma}(x_*) = \underbrace{K(x_*, x_*)}_{\text{prior variance}} - \underbrace{K(x_*, X) K(X, X)^{-1}K(X, x_*)}_{\text{variance modeled by IND set}}, \tag{24}$$

where $K(X, X) \in R^{N \times N}$ and $K(x_*, x_*) \in R$ denotes the kernel Gram matrix computed on the training inputs $X$, and the predictive input $x_*$, respectively.

We note that the kernel smoother employs the distance between the predictive input $x_*$ and the training (IND) set $X$ to model the predictive mean $\boldsymbol{\mu}(x_*)$ and variance $\boldsymbol{\Sigma}(x_*)$ in Eq. (23).

Using this observation, we first construct the smoother $\widehat{h}(x_*)$ by using the statistics of hidden feature $\{m_k\}_{k=1}^Q$, obtained from the pre-trained epoch $\mathcal{T}$, and $w_k(x)$ that inherently recognizes the distance of the hidden feature of $x_*$ from the features of IND set, as follows:

$$\widehat{h}(x_*) = \sum_{k=1}^Q w_k(x_*)\, m_k \in \mathbb{R}^H, \qquad w_k(x_*) = \frac{\exp(-\|\Delta_k(x_*)\|_{S^{-1}}^2)}{\sum_{j=1}^Q \exp(-\|\Delta_j(x_*)\|_{S^{-1}}^2)}. \tag{25}$$

Then, as we use the linear function $g(x) : R^H \longrightarrow R^Q$, defined as

$$g(x) = \theta^{(L)}\widehat{h}(x), \qquad \theta^{(L)} \sim \mathcal{N}\left(\theta^{(L)}; \widehat{\mu}, \text{diag}\left((\widehat{\sigma}_k^2)_{k=1}^Q\right)\right),$$

we design the mean of the function-space prior $\boldsymbol{\mu}(x_*)$ as

$$\boldsymbol{\mu}(x_*) = \text{E}[\, g(x_*)\,] = \widehat{\mu}\,\widehat{h}(x_*), \tag{26}$$

where $\widehat{h}(x_*)$ is considered to work similarly with the kernel smoother of the predictive mean in Eq. (23). Similarly, we design the variance of the function-space prior $\boldsymbol{\Sigma}(x_*)$ as

$$\boldsymbol{\Sigma}(x_*) = \text{diag}\left(\left(\, \underbrace{2\,\|m_{q_{x_*}}\|_{\widehat{\sigma}_k^2}^2}_{\text{SGD Prior}} - \underbrace{\|\widehat{h}(x_*)\|_{\widehat{\sigma}_k^2}^2}_{\text{Cov}[g(x_*)]_k}\,\right)_{k=1}^Q\right), \qquad q_{x_*} := \underset{k \in \{1,..,Q\}}{\arg\max}\, w_k(x_*), \tag{27}$$

where the SGD prior in Eq. (27) corresponds to the role of the prior variance of $K(x_*, x_*)$ in Eq. (24). The $\text{Cov}[g(x_*)]_k$ in Eq. (27) corresponds to the role of the variance modeled by IND set $K(x_*, X) K(X, X)^{-1}K(X, x_*)$ in Eq. (24).

A.4   PROOF OF PROPOSITION 4.1

In this section, we first present the Lemmas A.3 and A.4, and provide the proof of Proposition 4.1.

**Lemma A.3.** *For an input $x \in \mathcal{X}$ and the last-layer feature $h(x) := (f^{(L-1)} \circ \cdots \circ f^1)(x) \in R^H$, let $m_{-q} = \sum_{k \neq q} \frac{w_k(x)}{1 - w_q(x)} m_k$ and $\Delta m_q = m_{-q} - m_q$. Then, the $\widehat{h}(x)$ is re-expressed as follows:*

$$\widehat{h}(x) = m_q + (1 - w_q(x))\, \Delta m_q \tag{28}$$

*Proof.* In the following, we notate $h$ for $h(x)$ and $w_k$ for $w_k(x)$ for brevity. Then, the $\widehat{h}(x)$ is re-expressed as follows:

$$\widehat{h}(x) = \sum_{k=1}^{Q} w_k\, m_k = w_q m_q + (1 - w_q) \underbrace{\sum_{k \neq q} \frac{w_k}{1 - w_q} m_k}_{:= m_{-q}} = m_q + (1 - w_q) \big( \underbrace{m_{-q} - m_q}_{:= \Delta m_q} \big)$$

$\square$

**Lemma A.4.** *For $i, j \in \{1, .., Q\}$, suppose $\|m_i\|_2 = \|m_j\|_2$. Also for the parameter trajectory $\{\Theta(t)\}_{t \in \mathcal{T}}$, suppose that each element of $[\Theta^{(L)}(t)]_{k,h}$ is bounded by for some $0 < M < 1$, i.e., $\big|[\Theta^{(L)}(t)]_{k,h}\big| < M$ for $k \in \{1, .., Q\}$ and $h \in \{1, .., H\}$. Also, for each $k \in \{1, .., Q\}$, let us remind $\widehat{\sigma}_k^2$, defined as,*

$$\widehat{\sigma}_k^2 = \frac{1}{|\mathcal{T}|} \sum_{t \in \mathcal{T}} [\Theta^{(L)}(t)]_{k,:}^{\otimes 2} \quad - \quad [\widehat{\mu}_k]^{\otimes 2} \in \mathbb{R}_+^H, \tag{29}$$

*where $[\cdot]_{k,:}$ denotes $k$-th row and $\otimes 2$ denotes element-wise square. Then, following inequalities hold:*

$$(1)\ \|m_{-q}\|_2 \leq \|m_q\|_2, \qquad (2)\ \langle \Delta m_q\, , \, m_q \rangle < 0, \qquad (3)\ \langle \Delta m_q\, , \, m_q \rangle_{\widehat{\sigma}_k^2} < 0\ (w.h.p), \tag{30}$$

*Proof.* We use the same notation used in Lemma A.3.

The $(1)$ holds with the following reason:

$$\|m_{-q}\|_2 = \left\| \sum_{k \neq q} \frac{w_k}{1 - w_q} m_k \right\|_2 \leq \sum_{k \neq q} \frac{w_k}{1 - w_q} \|m_k\|_2 = \|m_q\|_2 \sum_{k \neq q} \frac{w_k}{1 - w_q} = \|m_q\|_2 \tag{31}$$

where the first inequality holds due to the triangle inequality, and second equality holds due to assumption $\|m_i\|_2 = \|m_j\|_2$.

The $(2)$ holds with the following reason:

$$\langle \Delta m_q\, , \, m_q \rangle = \langle\, m_{-q} - m_q\, , \, m_q\, \rangle = \|m_{-q}\|_2\, \|m_q\|_2 \cos\theta - \|m_q\|_2^2 \tag{32}$$

$$\leq \|m_q\|_2\, \|m_q\|_2 \cos\theta - \|m_q\|_2^2 \leq \|m_q\|_2^2 (\cos\theta - 1) \leq 0, \tag{33}$$

where the first inequality holds due to $(1)$. The last inequality holds only when the $m_q = m_k$ for $k \neq q$ because if there is some $k$ such that $m_k \neq m_q$, then $\cos(\theta_\angle) < 1$ for the angle $\theta_\angle$ between $m_q$ and $m_{-q}$.

The $(3)$ holds with the following reason:

Let $\widetilde{\sigma} = \big[ \widehat{\sigma}_k^2[1]\, , \, .. \, , \, \widehat{\sigma}_k^2[H] \big] \in R_+^H$ and $\widetilde{m} = \Delta m_q \circ m_q \in R^H$ for brevity; $\circ$ denotes the element-wise product. Then, $\langle \Delta m_q\, , \, m_q \rangle_{\widehat{\sigma}_k^2}$ can be re-expressed

$$\langle \Delta m_q\, , \, m_q \rangle_{\widehat{\sigma}_k^2} = \sum_{i=1}^{H} \widehat{\sigma}_k^2[i]\ (\Delta m_q[i]\, m_q[i]) = \langle \widetilde{\sigma}\, , \, \widetilde{m} \rangle, \tag{34}$$

where $\widehat{\sigma}_k^2[i]$, $\Delta m_q[i]$, and $m_q[i]$ denote the $i$-th element of each vector, respectively. Using the inner product in (2) can be re-expressed as $\langle \Delta m_q \, , \, m_q \rangle = \langle \mathbf{1}_H, \, \widetilde{m} \rangle$ with $\mathbf{1}_H = [1, .., 1] \in R^H$, $\langle \Delta m_q \, , \, m_q \rangle_{\widehat{\sigma}_k^2}$ can be also re-expressed

$$\langle \Delta m_q \, , \, m_q \rangle_{\widehat{\sigma}_k^2} = \langle \widetilde{\sigma} - \alpha \mathbf{1}_H \, , \, \widetilde{m} \rangle \, + \, \langle \alpha \mathbf{1}_H, \, \widetilde{m} \rangle \quad \text{for any } \alpha > 0. \tag{35}$$

Since $\langle \alpha \mathbf{1}_H \, , \, \widetilde{m} \rangle$ is a negative value due to result of (2), if $\langle \widetilde{\sigma} - \alpha \mathbf{1}_H \, , \, \widetilde{m} \rangle$ is proven to be much smaller value compared to $|\langle \alpha \mathbf{1}_H \, , \, \widetilde{m} \rangle|$, then $\langle \Delta m_q \, , \, m_q \rangle_{\widehat{\sigma}_k^2} < 0$ is also negative value. In this context, we proceed with this proof.

**Sub-Gaussian distribution of $\widetilde{\sigma}$.** To this end, we first show that each $\widetilde{\sigma}[h]$ is sub-Gaussian distribution; note $\widetilde{\sigma}[h] = \widehat{\sigma}_k^2[h]$. For $t \in \mathcal{T}$, let us assume each element of $t$-th trajectory weight parameter $\theta^{(L)}(t)$ is bounded by some $M > 0$, i.e., $|[\theta^{(L)}(t)]_{k,h}| < M$ for any $k \in \{1, .., Q\}$ and $h \in \{1, .., H\}$. Then, each element of the empirical variance $\widehat{\sigma}_k^2$ is bounded by $\frac{1}{|\mathcal{T}|}M^2$, as follows:

$$\widehat{\sigma}_k^2[h] = \frac{1}{|\mathcal{T}|} \sum_{t \in \mathcal{T}} [\Theta^{(L)}(t)]_{k,h}^{\otimes 2} \quad - \quad [\widehat{\mu}_k]_h^{\otimes 2} \leq \frac{1}{|\mathcal{T}|}M^2. \tag{36}$$

Then, we can regard $\widehat{\sigma}_k^2[h]$ as bounded random variable because $\widehat{\sigma}_k^2[h]$ could be different value depending on the parameter trajectory $\{\theta^{(L)}(t); t \in \mathcal{T}\}$ and $\widehat{\sigma}_k^2[h]$ is satisfied with $\widehat{\sigma}_k^2[h] \in \left[0, \frac{M^2}{|\mathcal{T}|}\right]$.

Then, since the bounded random variable $X \in [a, b]$ with zero mean is $\frac{(b-a)^2}{4}$ sub-Gaussian random variable due to Hoeffding's lemma (Van Handel, 2014), $\widehat{\sigma}_k^2[h] - \mathrm{E}[\widehat{\sigma}_k^2[h]]$ is also $\frac{M^4}{4|\mathcal{T}|^2}$ sub-Gaussian random variable.

**Mean of $\widetilde{\sigma}$.** Additionally, we assume $\mathrm{E}[\widehat{\sigma}_k^2[h_1]] = \mathrm{E}[\widehat{\sigma}_k^2[h_2]]$ for any $h_1, h_2 \in \{1, .., H\}$ and thus set $\alpha := \mathrm{E}[\widehat{\sigma}_k^2[h]]$. This is because each difference $\left| \mathrm{E}[\widehat{\sigma}_k^2[h_1]] - \mathrm{E}[\widehat{\sigma}_k^2[h_2]] \right|$ is bounded by $\frac{M^2}{|\mathcal{T}|}$ and thus would be small value if $M$ is small value such as $M < 1$.

**Concentration inequality** Next, using the Chernoff bound of the sub-Gaussian distribution (Zhang & Chen, 2020), we show that the tail probability of $\{\widetilde{\sigma}; \langle \widetilde{\sigma} - \alpha \mathbf{1}_H \, , \, \widetilde{m} \rangle > \epsilon\}$ is bounded as follows:

$$\mathrm{Pr}\left(\{\widetilde{\sigma}; \langle \widetilde{\sigma} - \alpha \mathbf{1}_H \, , \, \widetilde{m} \rangle > \epsilon\}\right) \leq \inf_{\lambda > 0} \exp\left(-\lambda \epsilon\right) \mathrm{E}\left[\exp\left(\langle \widetilde{\sigma} - \alpha \mathbf{1}_H \, , \, \lambda \widetilde{m} \rangle\right)\right] \tag{37}$$

$$= \inf_{\lambda > 0} \exp\left(-\lambda \epsilon\right) \prod_{h=1}^{H} \exp\left(\frac{\lambda^2 (\widetilde{m}[h])^2}{2} \frac{M^4}{4|\mathcal{T}|^2}\right) \tag{38}$$

$$\leq \inf_{\lambda > 0} \mathrm{E}\left[\exp\left(-\lambda \epsilon + \frac{\lambda^2}{2} \frac{\|\widetilde{m}\|_2^2 M^4}{4|\mathcal{T}|^2}\right)\right] = \exp\left(\frac{-2|\mathcal{T}|^2 \epsilon^2}{\|\widetilde{m}\|_2^2 M^4}\right) \tag{39}$$

This implies that with probability $1 - \delta$, the following inequality holds

$$\langle \widetilde{\sigma} \, , \, \widetilde{m} \rangle \, \leq \, \langle \alpha \mathbf{1}_H \, , \, \widetilde{m} \rangle \, + \, \frac{1}{\sqrt{2}} \log(\frac{1}{\delta}) \frac{\|\widetilde{m}\|_2 M^2}{|\mathcal{T}|}. \tag{40}$$

As we consider $\langle \alpha \mathbf{1}_H \, , \, \widetilde{m} \rangle = \alpha \sqrt{H} \|\widetilde{m}\|_2 \cos(\theta_\angle)$ with $\cos(\theta_\angle) < 0$ due to the result of (2) and $\alpha = \mathrm{E}[\widehat{\sigma}_k^2[h]] = C\frac{M^2}{|\mathcal{T}|}$ for some $C \in (0, 1)$, if the feature dimension $H$ is large enough to satisfy $H \geq \frac{(\log(\frac{1}{\delta}))^2}{2C^2 \cos^2(\theta_\angle)}$, then the right side of Eq. (40) would be negative for the following reason:

$$\langle \alpha \mathbf{1}_H \, , \, \widetilde{m} \rangle \, + \, \frac{1}{\sqrt{2}} \log(\frac{1}{\delta}) \frac{\|\widetilde{m}\|_2 M^2}{|\mathcal{T}|} = \Big( \underbrace{\sqrt{H} C \cos(\theta_\angle) + \frac{1}{\sqrt{2}} \log(\frac{1}{\delta})}_{<0 \text{ for large } H} \Big) \frac{\|\widetilde{m}\|_2 M^2}{|\mathcal{T}|} < 0. \tag{41}$$

Therefore, if each element of the weight parameter $\theta^{(L)}(t) \in \mathbb{R}^{Q \times H}$ is bounded by a small value $M$, and the feature dimension $H$ is large enough, then $\langle \widetilde{\sigma}, \widetilde{m} \rangle < 0$ holds with high probability. Note that the condition of $M$ and $H$ is easily feasible for the DNN.

$$\square$$

**Assumption.** Let assume that $\{m_q\}_{q=1}^{Q}$ and $\{\sigma_k\}_{k=1}^{Q}$ follow assumptions in Lemmas A.3 and A.4.

For $H$ feature dimension, let $H$ be large enough to satisfy $H \geq \mathcal{O}((\log(\frac{1}{\delta}))^2 \frac{1}{cos^2(\theta_\angle)})$ for small $\delta > 0$ and the angle $\theta_\angle$ between $\mathbf{1}_H = [1, .., 1] \in R^H$ and $\tilde{m}$ satisfying $\langle \Delta m_q , m_q \rangle = \langle \mathbf{1}_H, \tilde{m} \rangle$.

**Proposition A.5.** *For two input $x_1, x_2 \in \mathcal{X}$ and features $\widehat{h}(x_1), \widehat{h}(x_2) \in R^H$, let $k = q_{x_1} = q_{x_2}$ for some $k = \{1, .., Q\}$ meaning $m_k$ is their vicinity feature. Then, if $\widehat{h}(x_1)$ is not equal to but closer to $m_k$ than $\widehat{h}(x_2)$ in terms of MHD, i.e, $a_k < w_{q_{x_2}} < w_{q_{x_1}} < 1$ for*

$$a_k = \sup_{\{x \in \mathcal{X} \mid q_x = k\}} a(x) \quad \text{with} \quad a(x) = \max_{j \in \{1, .., Q\}} \frac{\langle m_{q_x}, m_{-q_x} \rangle_{\widehat{\sigma}_j^2}}{\|\Delta m_{q_x}\|_{\widehat{\sigma}_j^2}^2},$$

*then each $i$-th variance of $\mathbf{\Sigma}(x_1)$ is larger than that of $\mathbf{\Sigma}(m_k)$ and smaller than that of $\mathbf{\Sigma}(x_2)$,*

$$[\mathbf{\Sigma}(m_k)]_i < [\mathbf{\Sigma}(x_1)]_i < [\mathbf{\Sigma}(x_2)]_i \quad \text{for} \quad i = 1, .., Q, \tag{42}$$

*Proof.* For an input $x_1 \in \mathcal{X}$, let us assume $k = q_{x_1}$ with $q_{x_1} = 1$. Then, we can easily show $\widehat{h}(x_1) = m_k$ due to Eq. (10) and

$$\mathbf{\Sigma}(x_1) = \text{diag}\left(\left(2\|m_k\|_{\widehat{\sigma}_i^2}^2 - \|m_k\|_{\widehat{\sigma}_i^2}^2\right)_{i=1}^{Q}\right) = \text{diag}\left(\left(\|m_k\|_{\widehat{\sigma}_i^2}^2\right)_{i=1}^{Q}\right).$$

Next, for an input $x_2 \in \mathcal{X}$ satisfying $k = q_{x_2}$, we assume $w_{q_{x_2}} < w_{q_{x_1}} < 1$ intuitively meaning that $\widehat{h}(x_1)$ is closer to $m_k$ than $\widehat{h}(x_2)$ in sense of MHD. We show that each $k$-th component of the variance

$$[\mathbf{\Sigma}(x)]_k = 2\|m_{q_x}\|_{\widehat{\sigma}_k^2}^2 - \|\widehat{h}(x)\|_{\widehat{\sigma}_k^2}^2 = \|m_{q_x}\|_{\widehat{\sigma}_k^2}^2 + \underbrace{\|m_{q_x}\|_{\widehat{\sigma}_k^2}^2 - \|\widehat{h}(x)\|_{\widehat{\sigma}_k^2}^2}_{:=\rho_k(x)}$$

is an increasing function of $w_{q_x}$ on some range. This is because $\|m_{q_x}\|_{\widehat{\sigma}_k^2}^2$ is constant for given $q_x$ and $\rho_k(x)$ is an increasing function of $w_{q_x}$ as $w_{q_x}$ decreases from 1 to some constant $a \in (0, 1)$. To prove this statement, we will show that $\rho_k(x)$ satisfies the following properties for each $k = 1, ..Q$:

(1) $\rho_k(x) = 0$ for $w_{q_x} = 1$,

(2) $\rho_k(x)$ increases if $w_{q_x} \in \left(\frac{\langle m_{q_x}, m_{-q_x}\rangle_{\widehat{\sigma}_k^2}}{\|\Delta m_{q_x}\|_{\widehat{\sigma}_k^2}^2}, 1\right)$ moves from 1 to $\frac{\langle m_{q_x}, m_{-q_x}\rangle_{\widehat{\sigma}_k^2}}{\|\Delta m_{q_x}\|_{\widehat{\sigma}_k^2}^2}$,

To prove these properties, we first compute $\|\widehat{h}(x)\|_{\widehat{\sigma}_k^2}^2$, as follows:

$$\|\widehat{h}(x)\|_{\widehat{\sigma}_k^2}^2 = \|m_{q_x} + (1 - w_{q_x})\Delta m_q\|_{\widehat{\sigma}_k^2}^2 \tag{43}$$

$$= \|m_{q_x}\|_{\widehat{\sigma}_k^2}^2 + (1 - w_{q_x})^2 \|\Delta m_{q_x}\|_{\widehat{\sigma}_k^2}^2 + 2(1 - w_{q_x})\langle m_{q_x}, \Delta m_{q_x}\rangle_{\widehat{\sigma}_k^2}, \tag{44}$$

where the first equality holds due to Lemma A.3. Then, we can re-express $\rho_k(x)$ as follows:

$$\rho_k(x) = \|m_{q_x}\|_{\widehat{\sigma}_k^2}^2 - \|\widehat{h}(x)\|_{\widehat{\sigma}_k^2}^2 = -\left((1 - w_{q_x})^2 \|\Delta m_{q_x}\|_{\widehat{\sigma}_k^2}^2 + 2(1 - w_{q_x})\langle m_{q_x}, \Delta m_{q_x}\rangle_{\widehat{\sigma}_k^2}\right) \tag{45}$$

For the property of (1), we can easily show $p_k(x) = 0$ if we consider $w_{q_x} = 1$ for $p_k(x)$. To prove the property of (2), let us denote $b_q = 1 - w_{q_x} \in [0, 1)$ for brevity. Then, $\rho_k(x)$ is expressed as a second-order polynomial function of $b_q$ (concave), as follows:

$$\rho_k(x) = -\|m_{q_x}\|_{\sigma_k^2}^2 \underbrace{\left(b_q + \frac{\langle m_{q_x}, \Delta m_{q_x}\rangle_{\widehat{\sigma}_k^2}}{\|\Delta m_{q_x}\|_{\widehat{\sigma}_k^2}^2}\right)^2}_{<0} + \frac{(\langle m_{q_x}, \Delta m_{q_x}\rangle_{\widehat{\sigma}_k^2})^2}{\|\Delta m_{q_x}\|_{\widehat{\sigma}_k^2}^2} \tag{46}$$

$$= -\|m_{q_x}\|_{\sigma_k^2}^2 \left(w_{q_x} - \frac{\langle m_{q_x}, m_{-q_x}\rangle_{\widehat{\sigma}_k^2}}{\|\Delta m_{q_x}\|_{\widehat{\sigma}_k^2}^2}\right)^2 + \frac{(\langle m_{q_x}, \Delta m_{q_x}\rangle_{\widehat{\sigma}_k^2})^2}{\|\Delta m_{q_x}\|_{\widehat{\sigma}_k^2}^2} \tag{47}$$

where the inequality $\frac{\langle \mathrm{m}_{\mathrm{q_x}}, \Delta\mathrm{m}_{\mathrm{q_x}}\rangle_{\hat{\sigma}_k^2}}{\|\Delta\mathrm{m}_{\mathrm{q_x}}\|_{\hat{\sigma}_k^2}^2} < 0$ holds due to (3) in Lemma A.4.

The second equality holds due to $1 + \frac{\langle \mathrm{m}_{\mathrm{q_x}}, \Delta\mathrm{m}_{\mathrm{q_x}}\rangle_{\hat{\sigma}_k^2}}{\|\Delta\mathrm{m}_{\mathrm{q_x}}\|_{\hat{\sigma}_k^2}^2} = \frac{\langle \mathrm{m}_{\mathrm{q_x}}, \mathrm{m}_{\mathrm{-q_x}}\rangle_{\hat{\sigma}_k^2}}{\|\Delta\mathrm{m}_{\mathrm{q_x}}\|_{\hat{\sigma}_k^2}^2} < 1$. Then, since $\rho_k(x)$ is a

concave function having the maximum at $\frac{\langle \mathrm{m}_{\mathrm{q_x}}, \mathrm{m}_{\mathrm{-q_x}}\rangle_{\hat{\sigma}_k^2}}{\|\Delta\mathrm{m}_{\mathrm{q_x}}\|_{\hat{\sigma}_k^2}^2} < 1$, and $\rho_k(x) = 0$ for $w_{q_x} = 1$, $p_k(x)$

increases if $w_{q_x} \in \left( \frac{\langle \mathrm{m}_{\mathrm{q_x}}, \mathrm{m}_{\mathrm{-q_x}}\rangle_{\hat{\sigma}_k^2}}{\|\Delta\mathrm{m}_{\mathrm{q_x}}\|_{\hat{\sigma}_k^2}^2}, 1 \right)$ moves from 1 to $\frac{\langle \mathrm{m}_{\mathrm{q_x}}, \mathrm{m}_{\mathrm{-q_x}}\rangle_{\hat{\sigma}_k^2}}{\|\Delta\mathrm{m}_{\mathrm{q_x}}\|_{\hat{\sigma}_k^2}^2}$.

Then, the $\rho_k(x)$ is an increasing function of $w_{q_x}$ for all $k \in \{1, .., Q\}$ if $w_{q_x}(x)$ decreases in range of

$$w_{q_x}(x) \in \bigcap_{k=1}^{Q} \left( \frac{\langle \mathrm{m}_{\mathrm{q_x}}, \mathrm{m}_{\mathrm{-q_x}}\rangle_{\hat{\sigma}_k^2}}{\|\Delta\mathrm{m}_{\mathrm{q_x}}\|_{\hat{\sigma}_k^2}^2}, 1 \right] = \left( \underbrace{\max_k \frac{\langle \mathrm{m}_{\mathrm{q_x}}, \mathrm{m}_{\mathrm{-q_x}}\rangle_{\hat{\sigma}_k^2}}{\|\Delta\mathrm{m}_{\mathrm{q_x}}\|_{\hat{\sigma}_k^2}^2}}_{:=a(x)}, 1 \right]. \tag{48}$$

Therefore, each component of $\Sigma(x)$ is an increasing function of $w_{q_x}$ on this range of $w_{q_x}$ as well.

**Proof of the main statement** For $x_1, x_2 \in \mathcal{X}$ with $k = q_{x_1} = q_{x_2}$, we first consider $a_k = \sup_{\{x \in \mathcal{X} \mid q_x = k\}} a(x)$ using the $a(x)$ in Eq. (48). Then, if $a_k \leq w_{q_{x_2}} < w_{q_{x_1}} < 1$ intuitively meaning that $\hat{h}(x_1)$ is not equal to but closer to $m_k$ than $\hat{h}(x_2)$ in sense of MHD, the $i$-th diagonal variance of $\Sigma(x_1)$ is larger than that of Eq. (6) and smaller than $\Sigma(x_2)$, i.e.,

$$[\Sigma(m_k)]_i < [\Sigma(x_1)]_i < [\Sigma(x_2)]_i \qquad for \ i = 1, .., Q. \tag{49}$$

because $[\Sigma(x)]_i$ is an increasing function as $w_{q_x}$ decreases for all $x \in \{x \in \mathcal{X} \mid q_x = k\}$.

□

**Lemma A.6.** *(Analysis of predictive mean for classification) For $Q$-class classification task, let us assume that $q = \arg\max_{k \in \{1, .., Q\}} \langle \hat{\mu}_k, m_q \rangle$, meaning that $q$-th weight vector $\hat{\mu}_q$ leads the highest logits value for $q$-th feature $m_q$, where $\hat{\mu}$ is represented as*

$$\hat{\mu} = \frac{1}{|\mathcal{T}|} \sum_{t \in \mathcal{T}} \theta^{(L)}(t) \in \mathbb{R}^{Q \times H}. \tag{50}$$

*Then, the following inequality $[\mu(x_2)]_q < [\mu(x_1)]_q < [\mu(\mu_q)]_q$ holds where $[\mu(x)]_q$ denotes $q$-th logit (peaked) value of $\mu(x) \in R^Q$ in Eq. (6).*

*Proof.* For an input $x \in \mathcal{X}$, let us consider $q_x = \arg\max_{k=1}^{Q} w_k(x)$. Then, we show that $\mu(x)_{q_x}$

decreases as $w_{q_x}$ decreases for $w_{q_x}(x) \in \left( \max_k \frac{\langle \mathrm{m}_{\mathrm{q_x}}, \mathrm{m}_{\mathrm{-q_x}}\rangle_{\hat{\sigma}_k^2}}{\|\Delta\mathrm{m}_{\mathrm{q_x}}\|_{\hat{\sigma}_k^2}^2}, 1 \right]$ with following reason:

$$\mu(x)_{q_x} = \langle \hat{\mu}_{q_x}, (m_{q_x} + (1 - w_{q_x}) \Delta m_{q_x}) \rangle = \langle \hat{\mu}_{q_x}, m_{q_x} \rangle + (1 - w_{q_x}) \underbrace{\langle \mu_{q_x}, \Delta m_{q_x} \rangle}_{\leq 0},$$

where $\langle \hat{\mu}_{q_x}, \Delta m_{q_x} \rangle \leq 0$ holds with the following reason:

$$\langle \hat{\mu}_{q_x}, \Delta m_{q_x} \rangle = \sum_{k \neq q} \frac{w_k}{1 - w_{q_x}} \langle \hat{\mu}_{q_x}, m_k \rangle - \langle \hat{\mu}_{q_x}, m_{q_x} \rangle \tag{51}$$

$$= \sum_{k \neq q_x} \frac{w_k}{1 - w_{q_x}} \left( \underbrace{\langle \hat{\mu}_{q_x}, m_k \rangle - \langle \hat{\mu}_{q_x}, m_{q_x} \rangle}_{\leq 0 \text{ due to assumption}} \right) \leq 0, \tag{52}$$

□

## A.5 EXTENSION FOR REGRESSION TASK

We consider the following modifications for the regression, assuming a 1-dimensional function space ($Q = 1$) for brevity.

**Pseudo-label for MHD.** The MHD cannot be directly used for regression task because the MHD is defined using the discrete-valued label. Thus, we introduce the discrete pseudo label that is transformed by the real-valued output. To this end, for a continuous-valued label $Y = \{y_i\}_{i=1}^N$ in training set, we consider the range of $(-\infty, \min(Y)) \cup [\min(Y), \max(Y)] \cup (\max(Y), \infty)$, and partition this range into $K$ ordered intervals $\{\text{Bin}_k\}_{k=1}^K$, with $\text{Bin}_1 = [-\infty, \min(Y)), \text{Bin}_K = (\max(Y), \infty)$, and

$$\bigcup_{k=1}^K \text{Bin}_k = (-\infty, \min(Y)) \cup [\min(Y), \max(Y)] \cup (\max(Y), \infty). \tag{53}$$

Then, we assign the pseudo label $L(y_i) := k$ if $y_i \in \text{Bin}_k$. For the tuple of $(x_i, y_i, L(y_i))$ with $L(y_i) \in \{1, .., K\}$, we compute $m_k$ and $S$ in Eq. (7) using $L(y_i)$ instead of $y_i$ with $N_k = |\{i \mid L(y_i) = k\}|$, as follows:

$$m_k = \frac{1}{N_k} \sum_{i: L(y_i)=k} h(x_i), \quad S = \frac{1}{N} \sum_{k=1}^Q \sum_{i: L(y_i)=k} \Delta_k(x_i), \quad \Delta_k(x) = h(x) - m_k$$

**Variance of the function-space prior.** The covariance $\Sigma(x)$ of Eq. (6) using the pseudo label, consists of $K \times K$ diagonal covariance representing the variances of $K$ intervals in function space. This $K$ could be different to the dimension of the output ($Q = 1$).

Thus, we consider to choose the variance of the specific interval using $q_x = \arg\max_{k \in \{1,..,K\}} w_k(x)$, and define the one-dimensional variance $\Sigma(x) \in R_+$ as follows:

$$\Sigma(x) = 2 \underbrace{\|m_{q_x}\|_{\widehat{\sigma}^2}^2}_{\text{SGD Prior}} - \underbrace{\|\widehat{h}(x)\|_{\widehat{\sigma}^2}^2}_{\text{Var}[g(x)]} \quad with \quad q_x = \arg\max_{k \in \{1,..,K\}} w_k(x),$$

where $g(x) = \theta^{(L)} \widehat{h}(x)$ using the projected feature $\widehat{h}(x)$ of Eq. (10) and the last weight random weight parameter $\theta^{(L)} \sim \mathcal{N}(\theta^{(L)}; \widehat{\mu}, \widehat{\sigma}^2)$. The average of mean $\widehat{\mu} \in R^H$ and standard deviation $\widehat{\sigma}^2 \in R_+^H$ are obtained by Eq. (9) for 1-D regression. This can be naturally extended for $Q$-D regression by using $\widehat{\mu} \in R^Q$ and $\text{diag}((\widehat{\sigma}_q^2)_{q=1}^Q) \in R_+^{Q \times Q}$ for last weight random parameter $\theta^{(L)}$.

## B    APPENDIX:EXPERIMENT DETAILS

### B.1    ADDITIONAL EXPERIMENT RESULTS FOR SECTION 5.1

**Experiment setting.**    We follow the established training hyperparameter configurations as outlined in He et al. (2016). For ResNet 20 training on CIFAR 10, we use 200 training epochs, a batch size of 128, and use the SGD optimizer with a learning rate of 0.1, weight decay of $5 \times 10^{-4}$, and momentum of 0.9. The cosine learning scheduler is applied after 10 warm-up epochs.

Additionally, we introduce the scale hyperparameter to increase the variance of the weight-space prior $\widehat{\sigma}_k^2$ in Eq. (9) because the variance of the weight-space prior obtained from SGD trajectory is often too small, potentially leading to numerical errors. We also consider to constrain the dimension of the function-space prior by selecting the top-k dimensions of the function-output based on the mean parameters $\boldsymbol{\mu}(x)$ of the function-space prior in Eq. (6). Subsequently, we apply KL regularization to the constrained dimension in function space.

The other configurations of the inference method is described in Table 3.

| Inference | Hyperparameters | Range |
|---|---|---|
| T-FVI, R-FVI | KL regularization (relative) $\lambda$ in Eq. (1) | $\{10^{-1}, 10^{-2}, 10^{-3}\}$ |
| T-FVI, R-FVI | Variance of of variational weight parameters (log) | $\mathcal{U}(-6, -5)$ |
| T-FVI, R-FVI | The number of context inputs per batch | 32 / 128 |
| R-FVI | Pre-determined iterations $\mathcal{T}$ | $\mathcal{T}_{\mathrm{ResNet}}$ |
| R-FVI | Radius $r$ in Eq. (13) for adversarial feature | $\{0.05, 0.10, 0.20\}$ |
| R-FVI | Scale of the variance of weight-space prior $\widehat{\sigma}_k^2$ | 10 |
| R-FVI | Restriction of function-space prior (TopK) | 3 (CIFAR 10) |

Table 3: Hyperparameters settings of the proposed inference (R-FVI)

For computational resource, we used an RTX 2080 (11 GB) to run experiments.

### B.1.1    INVESTIGATION OF THE FUNCTION-SPACE PRIOR CONSTRUCTED BY DIFFERENT SGD TRAJECTORIES.

Following the experiment setting in Section 6.1, we further investigate the function-space prior constructed by different SGD trajectories. For training epoch $T = 200$, we consider the SGD trajectories $\mathcal{T}_{\mathrm{ResNet}} = \{\mathcal{T}_1, \mathcal{T}_2, \mathcal{T}_3, \mathcal{T}_4\}$ where each $\mathcal{T}_i$ for $i = 1, 2, 3, 4$, is defined as follows:

$\mathcal{T}_1 = \{0.50T - 20, 0.50T - 16, 0.50T - 12, 0.50T - 8, 0.50T - 4\}$,
$\mathcal{T}_2 = \{0.70T - 20, 0.70T - 16, 0.70T - 12, 0.70T - 8, 0.70T - 4\}$,
$\mathcal{T}_3 = \{0.80T - 20, 0.80T - 16, 0.80T - 12, 0.80T - 8, 0.80T - 4\}$,
$\mathcal{T}_4 = \{0.80T - 10, 0.80T - 8, 0.80T - 6, 0.80T - 4, 0.80T - 2\}$.

Fig. 8 shows the averaged $w_{q_x}$ of Eq. (10) over IND set (CIFAR 10), OOD set (SVHN), and the adversarial hidden feature $z_{\text{adv}}$ of Eq. (13) with radius $r \in \{.05, .10, .20\}$ for the function-space priors constructed by SGD trajectories $\{\mathcal{T}_1, \mathcal{T}_2, \mathcal{T}_3, \mathcal{T}_4\}$. Fig. 9 shows the corresponding averaged standard deviation of the function-space priors, i.e, $\text{Tr}(\mathbf{\Sigma}^{\frac{1}{2}}(x))$ of Eq. (6), respectively. These figures imply that when the parameter trajectory of SGD iterations contains sufficient information to discern whether the feature of an input is likely to be an in-distribution (IND) feature, as illustrated in Figs. 8c and 8d, then their function-space priors constructed by $\mathcal{T}_3$ and $\mathcal{T}_4$ induce the larger levels of uncertainty into the model as the hidden feature $\widehat{h}$ is likely to be OOD set as shown in Figs. 9c and 9d. These results demonstrate our statements in Proposition 4.1 and Lemma 4.2.

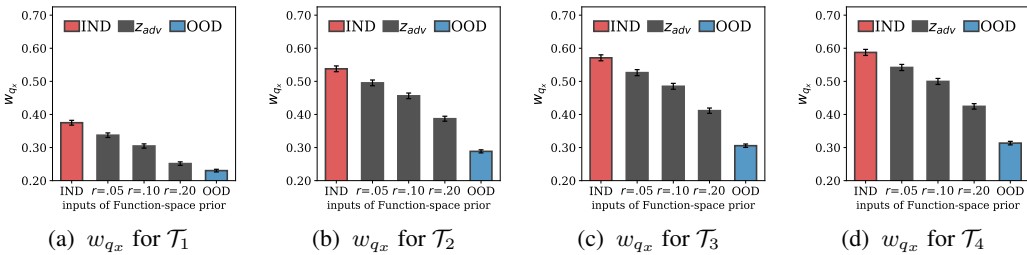

Figure 8: Investigation on $w_{q_x}$ using the different SGD trajectories $\{\mathcal{T}_1, \mathcal{T}_2, \mathcal{T}_3, \mathcal{T}_4\}$.

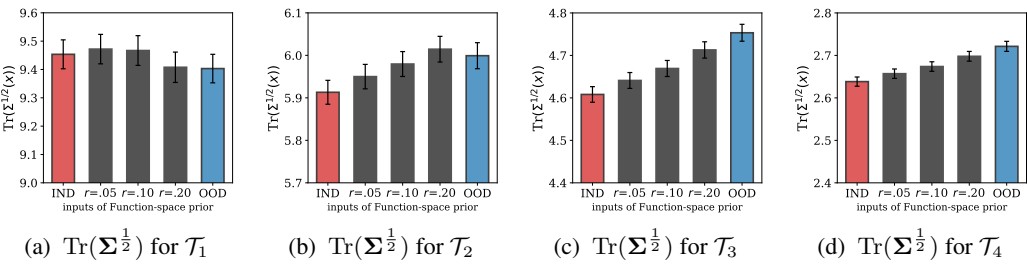

Figure 9: Investigation on $\text{Tr}(\mathbf{\Sigma}^{\frac{1}{2}})$ using the different SGD trajectories $\{\mathcal{T}_1, \mathcal{T}_2, \mathcal{T}_3, \mathcal{T}_4\}$.

**Comparison of the R-FVI and F-prior.** Fig. 10 compares the ACC, NLL (CIFAR 10), and AUROC (SVHN) of the R-FVIs (KL regularization hyperparameter $\lambda = 0.1$) and those of their function-space priors constructed by SGD trajectories $\{\mathcal{T}_1, \mathcal{T}_2, \mathcal{T}_3, \mathcal{T}_4\}$, respectively. We use the 10 predictive sample functions ($J = 10$) for Bayesian model averaging (BMA) prediction and obtain the results over 3 random seeds.

This figure shows that if the SGD trajectory is well selected like $\{\mathcal{T}_3, \mathcal{T}_4\}$, their corresponding function-space variational posterior leads to superior performance on IND set (higher ACC) and on OOD set (higher AUROC).

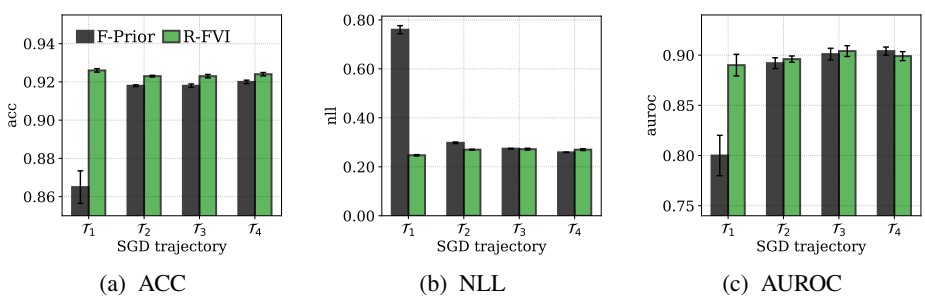

Figure 10: Investigation on the performances obtained from different SGD trajectories $\{\mathcal{T}_1, \mathcal{T}_2, \mathcal{T}_3, \mathcal{T}_4\}$.

### B.1.2 HOW DOES THE RELATIONSHIP BETWEEN CONTEXT AND TRAINING SET AFFECT T-FVI'S PERFORMANCE ?

Following the experiment setting in Section 6.1, we further investigate the effect of the context set on the performance of T-FVI using the uniform Gaussian function-space prior $\mathcal{N}(0, 10I_{Q \times Q})$; 10 is empirically found over $\{5, 10, 50\}$. We consider the context set

$$x_{\text{cxt}} = (1 - \alpha)x_{\text{tr}} + \alpha x_{\text{add}}$$

by introducing the external dataset $x_{\text{ext}}$ and then mixing $x_{\text{ext}}$ with training set $x_{\text{tr}}$ with the mixing level $\alpha \in (0, 1)$; if $\alpha$ is close to 0, the context set can be regarded as the IND-context set close to $x_{\text{tr}}$.

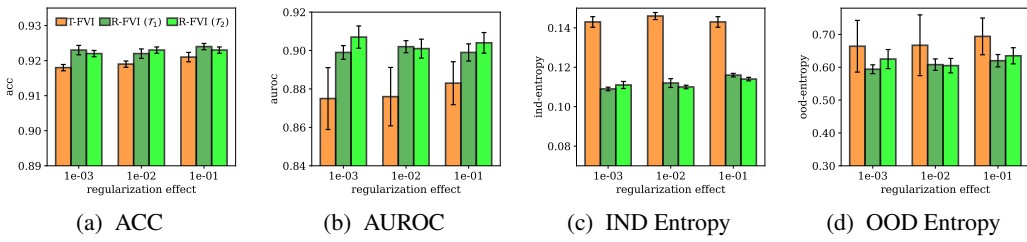

(a) ACC      (b) AUROC      (c) IND Entropy      (d) OOD Entropy

Figure 11: Performance comparison between T-FVI and R-FVIs using mixing level $\alpha = 0.2$

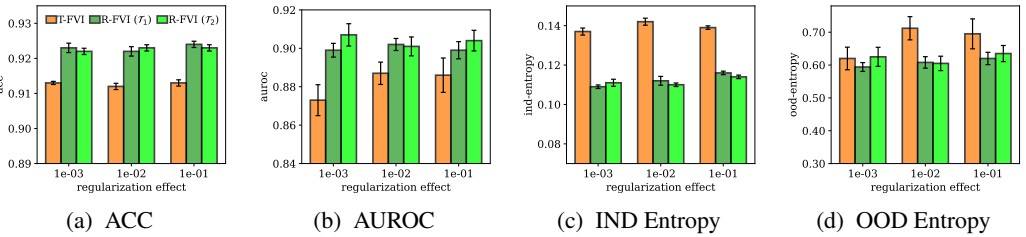

(a) ACC      (b) AUROC      (c) IND Entropy      (d) OOD Entropy

Figure 12: Performance comparison between T-FVI and R-FVIs using mixing level $\alpha = 1.0$

**Results.** Figs. 11 and 12 compare the results of the baseline inference (T-FVI) the proposed inference (R-FVI) using different SGD trajectories:

$$\mathcal{T}_1 = \{0.80T - 10, 0.80T - 8, 0.80T - 6, 0.80T - 4, 0.80T - 2\},$$
$$\mathcal{T}_2 = \{0.80T - 20, 0.80T - 16, 0.80T - 12, 0.80T - 8, 0.80T - 4\}$$

where $T = 200$, $x_{\text{tr}} = \text{CIFAR10}$, $x_{\text{add}} = \text{CIFAR100}$, and mixing level $\alpha \in \{0.2, 1.0\}$ are considered. The x-axis denotes the relative regularization hyperparameter $\lambda \in \{10^{-3}, 10^{-2}, 10^{-1}\}$ that applies the same amount of the regularization to the model as described in Section 6.1, and the y-axis denotes the corresponding metric.

Figs. 11 and 12 imply that when the context input is less likely to be the IND set ($\alpha = 1.0$), the performance of T-FVI on the IND set (ACC) degrades as shown in Figs. 11a and 12a, while its performance on the OOD set (AUROC) improves as shown in Figs. 11b and 12b. Notably, the predictive entropy of T-FVI on the IND set is consistently higher, as shown in Figs. 11c and 12c, whereas its predictive entropy on the OOD set increases when $\alpha = 1.0$, as shown in Figs. 11d and 12d.

Fig. 13 compares the predictive sample functions of our prior on IND set and OOD sets.

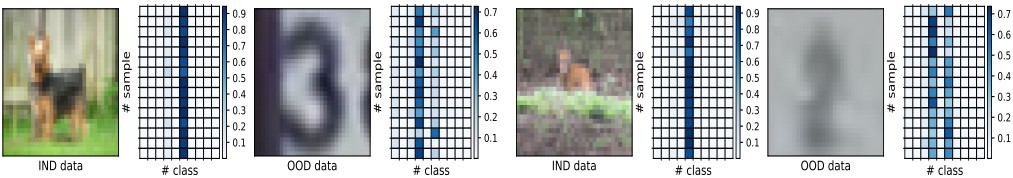

Figure 13: These figures describes predictive samples $\{\text{softmax}(f_i(x))\}_{i=1}^{15}$ of our prior for each two IND (CIFAR 10) and OOD (SVHN) data, implying our prior yields more uncertainty on OOD data.

## B.2 ADDITIONAL EXPERIMENT RESULTS FOR SECTION 5.2

**Experiment setting.** We follow the established training hyperparameter configurations as outlined in He et al. (2016). For ResNet 18 and 50 training on CIFAR-10 and CIFAR-100 respectively, we follow the same configuration of ResNet training on CIFAR 10, described in Appendix B.1.

We compare our method with the following baselines: Maximum a posterior (MAP), Stochastic weight averaging Gaussian (SWAG) Maddox et al. (2019), Spectral-normalized Gaussian process (SNGP) Liu et al. (2020b), Mean-field weight-space Variational inference Blundell et al. (2015) (WVI) using fully Bayesian layer (FL) and last Bayesian layer (LL), and T-FVI Rudner et al. (2022).

The other configurations are described in Table 4.

| Inference | Hyperparameters | Range |
|:---:|:---|:---:|
| MAP | Regularization $\lambda$ | $\{10^{-3}, 10^{-4}\}$ |
| T-FVI, R-FVI | KL regularization $\lambda$ in Eq. (1) | $\{10^{-3}, 10^{-4}, 10^{-5}\}$ |
| T-FVI, R-FVI | Variance of of variational weight parameters (log) | $\mathcal{U}(-6, -5)$ |
| T-FVI, R-FVI | The number of context inputs per batch | 32 / 128 |
| R-FVI | Pre-determined iterations $\mathcal{T}$ | $\mathcal{T}_{\text{ResNet}}$ |
| R-FVI | Radius $r$ in Eq. (13) for adversarial feature | $\{0.05, 0.10, 0.15\}$ |
| R-FVI | Scale of the variance of weight-space prior $\widehat{\sigma}_k^2$ | 10 |
| R-FVI | Restriction of function-space prior (TopK) | 3 (CIFAR 10), 10 (CIFAR 100) |

Table 4: Hyperparameters settings of the proposed inference (R-FVI)

For the R-FVI, we consider the following SGD trajectories $\mathcal{T}_{\text{ResNet}} = \{\mathcal{T}_1, \mathcal{T}_2, \mathcal{T}_3\}$ with $T = 200$:

$$\mathcal{T}_1 = \{0.75T - 20, 0.75T - 16, 0.75T - 12, 0.75T - 8, 0.75T - 4\},$$
$$\mathcal{T}_2 = \{0.80T - 20, 0.80T - 16, 0.80T - 12, 0.80T - 8, 0.80T - 4\},$$
$$\mathcal{T}_3 = \{0.85T - 20, 0.85T - 16, 0.85T - 12, 0.85T - 8, 0.85T - 4\},$$

For computational resource, we used RTX 2080 (11 GB) and RTX 3090 TI (24 GB).

### B.2.1 DEMONSTRATION OF PRIOR PROPERTY FOR CLASSIFICATION TASK

Furthermore, we empirically demonstrate the property of the function-space prior in Proposition 4.1 and Lemma 4.2 for ResNet 18 and 50. We use the trained models which are reported in Table 1. For comparison, we consider the random Gaussian perturbation of the last-layer hidden feature, i.e., $h + r$ with $r \sim \mathcal{N}(0, r^2)$ instead of using the adversarial hidden feature $z_{\text{adv}}$ using the radius $r$.

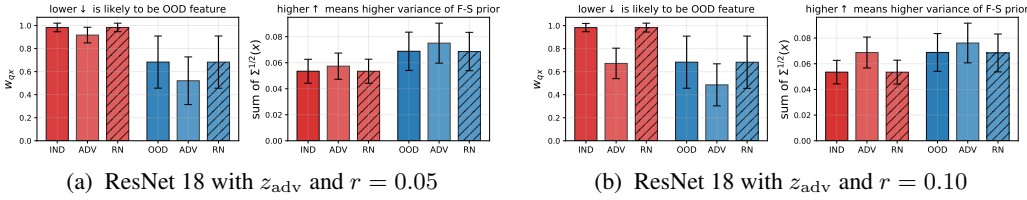

(a) ResNet 18 with $z_{\text{adv}}$ and $r = 0.05$      (b) ResNet 18 with $z_{\text{adv}}$ and $r = 0.10$

Figure 14: Demonstration of the property of the function-space prior in Proposition 4.1 and Lemma 4.2

Fig. 14a shows the result of ResNet 18 using the R-FVI with $r = 0.05$. The left panel shows $w_{q_x}$ with $q_x = \arg\max_{k=1}^Q w_k(x)$, evaluated on the IND set (CIFAR-10), the OOD set (SVHN), the adversarial hidden feature $z_{\text{adv}}$ from Eq. (13), and the random Gaussian perturbation (RN). The right panel shows the sum of the standard deviation $\text{Tr}(\boldsymbol{\Sigma}^{1/2}(x))$ of the function-space prior over each dataset. Similarly, Fig. 14b shows the corresponding results of using $r = 0.10$. Note that as $r$ increases from $r = 0.05$ to $r = 0.10$, the value of $w_{q_x}$ decreases and $\text{Tr}(\boldsymbol{\Sigma}^{1/2}(x))$ increases. Figs. 15a and 15b show the corresponding results of the ResNet 50 using R-FVI with $r = 0.10$ and 0.20, respectively.

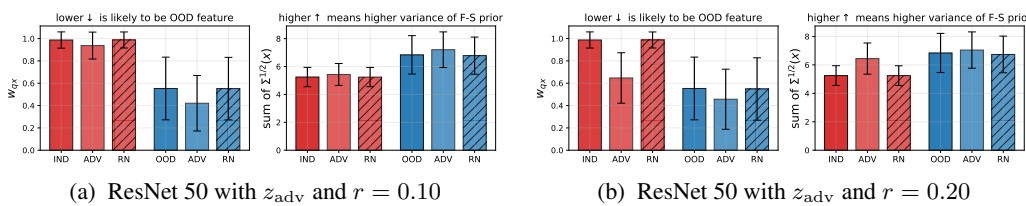

(a) ResNet 50 with $z_{\mathrm{adv}}$ and $r = 0.10$      (b) ResNet 50 with $z_{\mathrm{adv}}$ and $r = 0.20$

Figure 15: Demonstration of the property of the function-space prior in Proposition 4.1 and Lemma 4.2

From these figures, we confirm that the function-space prior of the trained model can assign the different levels of the uncertainty into the model depending on the status of the input, which is stated in Proposition 4.1 and Lemma 4.2. That is, as the inputs are less likely to come from the IND set, the value of $w_{q_x}$ decreases. The sum of the corresponding standard deviation of the function-space prior $\mathrm{Tr}(\boldsymbol{\Sigma}^{1/2}(x))$ increases as the value of $w_{q_x}$ decreases. This behavior is also observed for $z_{\mathrm{adv}}$, whereas the value of $w_{q_x}$ remains almost constant for RN.

**Qualitative analysis of the function-space prior.** We present examples of the random predictive probabilities ($J = 15$) of R-FVI and T-FVI, evaluated on IND and OOD set, in Fig. 16. This visualization shows that R-FVI leads to confident predictions on the IND set as well as inconsistent predictions on the OOD set as compared to those of T-FVI. This is possibly due to the KL regularization through the proposed function-space prior.

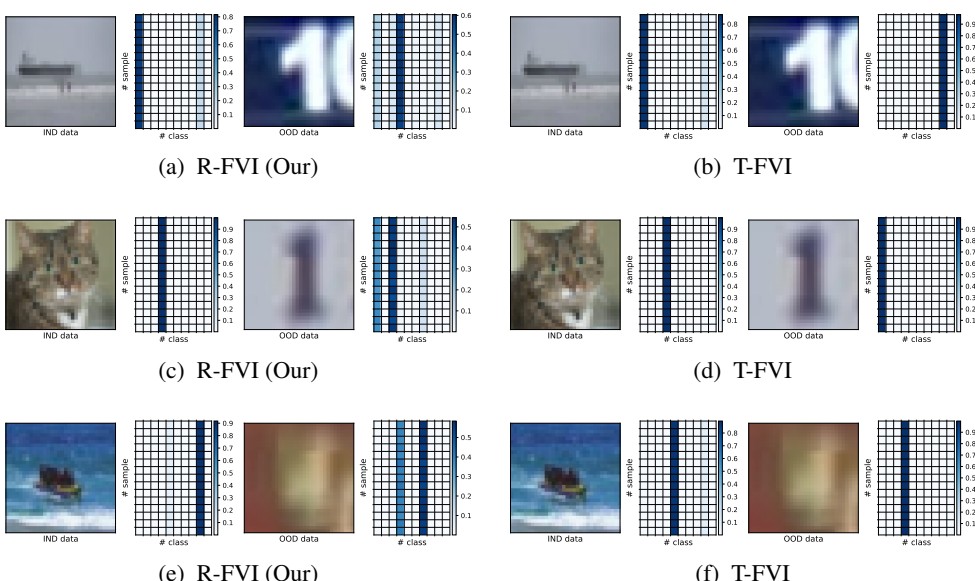

(a) R-FVI (Our)      (b) T-FVI

(c) R-FVI (Our)      (d) T-FVI

(e) R-FVI (Our)      (f) T-FVI

Figure 16: Comparison of 15 predictive sample probabilities for IND (CIFAR 10) and OOD (SVHN).

### B.2.2 INVESTIGATION OF THE EFFECT OF VARYING HYPERPARAMETERS ON R-FVI.

**Parameter trajectories of SGD iterations.** We first investigate how the parameter trajectory of SGD iterations affects the performance. We consider the setting of CIFAR 100 using ResNet 50.

We set the KL regularization hyperparameter $\lambda = 10^{-3}$, the scale of the variance of weight-space prior $S = 10$, the radius of adversarial hidden feature $r = 0.1$, the constrained dimension of the function output $\text{TopK} = 10$ for regularization as described in **experiment setting**. Then, we consider the following SGD trajectories with $T = 200$:

$\mathcal{T}_1 = \{0.75T - 20, 0.75T - 16, 0.75T - 12, 0.75T - 8, 0.75T - 4\}$,
$\mathcal{T}_2 = \{0.80T - 20, 0.80T - 16, 0.80T - 12, 0.80T - 8, 0.80T - 4\}$,
$\mathcal{T}_3 = \{0.85T - 20, 0.85T - 16, 0.85T - 12, 0.85T - 8, 0.85T - 4\}$,

Table 5 shows the results of the ResNet 50 trained by R-FVI using the parameter trajectories $\mathcal{T}_1, \mathcal{T}_2$, and $\mathcal{T}_3$. The R-FVI using the trajectories $\mathcal{T}_2$ and $\mathcal{T}_3$ improves the uncertainty estimation on IND set and OOD set compared to those of MAP.

| SGD Trajectory | # sample | ACC ↑ | NLL ↓ | ECE ↓ | AUROC ↑ |
|---|---|---|---|---|---|
| MAP | J=1 | (0.797, 0.015) | (0.835, 0.002) | (0.074, 0.002) | (0.807, 0.014) |
| R-FVI w. $\mathcal{T}_1$ | J=1 | (0.797, 0.005) | (0.835, 0.015) | (0.075, 0.001) | (0.827, 0.018) |
| | J=5 | (0.798, 0.005) | (0.820, 0.017) | (0.072, 0.002) | (0.829, 0.017) |
| | J=10 | (0.799, 0.005) | (0.819, 0.017) | (0.072, 0.001) | (0.829, 0.017) |
| R-FVI w. $\mathcal{T}_2$ | J=1 | (0.797, 0.005) | (0.819, 0.015) | (0.071, 0.002) | (0.843, 0.015) |
| | J=5 | (0.798, 0.006) | (0.815, 0.016) | (0.070, 0.002) | (0.844, 0.015) |
| | J=10 | (0.798, 0.006) | (0.815, 0.016) | (0.070, 0.002) | (0.844, 0.015) |
| **R-FVI w. $\mathcal{T}_3$** | J=1 | (0.800, 0.005) | (0.791, 0.011) | (0.062, 0.001) | (0.846, 0.010) |
| | J=5 | (0.802, 0.005) | (0.790, 0.012) | (0.061, 0.001) | (0.846, 0.010) |
| | J=10 | (0.801, 0.004) | (0.790, 0.011) | (0.061, 0.000) | (0.846, 0.010) |

Table 5: Investigation the performance for varying the parameter trajectories of the SGD iterations.

**Radius $r$ of the adversarial hidden feature.** We also investigate how the radius of the adversarial hidden feature $z_{\text{adv}}$ affects the performance. We set the trajectory $\mathcal{T}_3$ and consider the following radius $r$ as described in Table 6, where $\mathcal{U}(a, b)$ denotes uniform distribution defined on $[a, b]$.

From Table 6, we see that using the random perturbation on the radius $r \in \mathcal{U}(0.05, 0.15)$ can improve ECE evaluated on IND set and the AUROC evaluated on OOD set.

| Radius for $z_{\text{adv}}$ | # sample | ACC ↑ | NLL ↓ | ECE ↓ | AUROC ↑ |
|---|---|---|---|---|---|
| MAP | J=1 | (0.797, 0.015) | (0.835, 0.002) | (0.074, 0.002) | (0.807, 0.014) |
| $r = 0.10$ | J=1 | (0.800, 0.005) | (0.791, 0.011) | (0.062, 0.001) | (0.846, 0.010) |
| | J=5 | (0.802, 0.005) | (0.790, 0.012) | (0.061, 0.001) | (0.846, 0.010) |
| | J=10 | (0.801, 0.004) | (0.790, 0.011) | (0.061, 0.000) | (0.846, 0.010) |
| $r \in \mathcal{U}(0.05, 0.10)$ | J=1 | (0.802, 0.005) | (0.799, 0.014) | (0.063, 0.002) | (0.845, 0.012) |
| | J=5 | (0.802, 0.004) | (0.797, 0.014) | (0.063, 0.001) | (0.845, 0.012) |
| | J=10 | (0.801, 0.005) | (0.797, 0.014) | (0.062, 0.001) | (0.845, 0.012) |
| **$r \in \mathcal{U}(0.05, 0.15)$** | J=1 | (0.799, 0.004) | (0.794, 0.012) | (0.057, 0.001) | (0.849, 0.015) |
| | J=5 | (0.799, 0.003) | (0.792, 0.012) | (0.056, 0.000) | (0.850, 0.014) |
| | J=10 | (0.799, 0.003) | (0.792, 0.012) | (**0.056**, 0.000) | (**0.850**, 0.014) |
| $r \in \mathcal{U}(0.10, 0.15)$ | J=1 | (0.801, 0.005) | (0.790, 0.013) | (0.060, 0.001) | (0.847, 0.012) |
| | J=5 | (0.801, 0.004) | (0.789, 0.014) | (0.060, 0.001) | (0.847, 0.012) |
| | J=10 | (0.801, 0.004) | (0.789, 0.013) | (0.061, 0.001) | (0.847, 0.012) |

Table 6: Investigation the performance for varying the parameter trajectories of the SGD iterations.

**Comparison with the context feature as noise perturbation.** Following setting previous experiment for training ResNet 18 on CIFAR 10, we compare the R-FVI using the adversarial hidden

feature $z_{\mathrm{adv}}$ from Eq. (13) and the random Gaussian perturbation (RN) to investigate the effectiveness of the $z_{\mathrm{adv}}$. Additionally, we compare them with R-FVI using the only RP trick on function-space without using the function-space KL divergence regularization to investigate the effectiveness of the feature-distribution-aware prior.

| Method | # sample | ACC ↑ | NLL ↓ | ECE ↓ | AUROC ↑ |
|---|---|---|---|---|---|
| R-FVI w. $z_{\mathrm{adv}}$ ($r = .10$) | J=1 | (0.952, 0.001) | (0.187, 0.005) | (0.027, 0.002) | (0.955, 0.004) |
| | J=5 | (0.952, 0.001) | (0.187, 0.005) | (0.027, 0.002) | (0.956, 0.004) |
| | J=10 | (0.952, 0.001) | (0.186, 0.005) | (0.027, 0.001) | (**0.956**, 0.004) |
| R-FVI w. RN ($r = .10$) | J=1 | (0.952, 0.001) | (0.185, 0.003) | (0.026, 0.001) | (0.952, 0.009) |
| | J=5 | (0.952, 0.001) | (0.185, 0.003) | (0.025, 0.001) | (0.952, 0.009) |
| | J=10 | (0.952, 0.001) | (0.185, 0.003) | (0.026, 0.001) | (0.952, 0.009) |
| R-FVI w/o regularization | J=1 | (0.948, 0.001) | (0.199, 0.004) | (0.030, 0.001) | (0.940, 0.011) |
| | J=5 | (0.948, 0.002) | (0.199, 0.005) | (0.030, 0.001) | (0.940, 0.011) |
| | J=10 | (0.948, 0.002) | (0.199, 0.005) | (0.030, 0.001) | (0.940, 0.011) |

Table 7: Comparison of R-FVI using the adversarial feature $z_{\mathrm{adv}}$ and the random perturbation (RN).

Table 7 shows that using the proposed prior with $z_{\mathrm{adv}}$ and Gaussian perturbation (RN) leads to better uncertainty estimation on both IND set (higher NLL and ECE) and the OOD set (higher AUROC) than that of using on RP trick (w.o regularization). Also, this result implies that using $z_{\mathrm{adv}}$ leads to better uncertainty estimation on the OOD set (higher AUROC) than that of using Gaussian perturbation.

**Comparison with variants of T-FVI using non-linear layers.** We conduct additional experiments on CIFAR-10 using ResNet 18 to demonstrate that using the structure of the last-layer BNN with R-FVI is effective. To this end, we compare the proposed method with variants of T-FVI replacing the last linear layer ([512, 10] with 512 layer features and 10 classes) to the following layers:

T-FVI-2: a Bayesian 2-hidden MLP layer ([512, 128] $\rightarrow$ ReLU $\rightarrow$ [128, 10]), and

T-FVI-3: a Bayesian 3-hidden MLP layer ([512, 256] $\rightarrow$ ReLU $\rightarrow$ [256, 128] $\rightarrow$ ReLU $\rightarrow$ [128, 10]).

| Method | # sample | ACC ↑ | NLL ↓ | ECE ↓ | AUROC ↑ |
|---|---|---|---|---|---|
| T-FVI | J=1 | (0.943, 0.004) | (0.216, 0.011) | (0.032, 0.002) | (0.927, 0.009) |
| | J=5 | (0.943, 0.004) | (0.216, 0.011) | (0.032, 0.002) | (0.927, 0.009) |
| | J=10 | (0.943, 0.004) | (0.216, 0.011) | (0.032, 0.002) | (0.927, 0.009) |
| T-FVI-2 | J=1 | (0.945, 0.001) | (0.214, 0.006) | (0.032, 0.001) | (0.924, 0.012) |
| | J=5 | (0.945, 0.001) | (0.214, 0.006) | (0.032, 0.001) | (0.924, 0.013) |
| | J=10 | (0.945, 0.001) | (0.213, 0.006) | (0.032, 0.001) | (0.924, 0.013) |
| T-FVI-3 | J=1 | (0.946, 0.001) | (0.220, 0.006) | (0.031, 0.001) | (0.931, 0.007) |
| | J=5 | (0.947, 0.001) | (0.219, 0.005) | (0.030, 0.001) | (0.931, 0.007) |
| | J=10 | (0.946, 0.001) | (0.219, 0.005) | (0.031, 0.001) | (0.931, 0.007) |
| R-FVI | J=1 | (0.952, 0.001) | (0.187, 0.005) | (0.028, 0.002) | (0.956, 0.004) |
| | J=5 | (0.952, 0.001) | (0.187, 0.005) | (0.028, 0.002) | (0.956, 0.004) |
| | J=10 | (**0.952**, 0.001) | (**0.187**, 0.005) | (**0.028**, 0.001) | (**0.956**, 0.004) |

Table 8: Comparison with variants of T-FVI using non-linear layers on IND set (CIFAR 10) and OOD set (SVHN).

**Results.** Table 8 shows that R-FVI consistently outperforms the variants of the T-FVI using non-linear mapping that uses an increasing number of weight parameters for the mean and variance parameters of the weight-space variational and prior distribution. In addition, we attempted to compare higher-order MLP layers (4 - 10 layers) with dropout ($p = 0.5$), and observed that the models were significantly under-fitted. Therefore, we want to emphasize that this performance improvement of R-FVI is not marginal.

### B.2.3 COMPARISON WITH THE GAUSSIAN PROCESS (GP) LAST-LAYER

Following the hyperparameters of Wide-ResNet described in the appendix Liu et al. (2020b), we set the hyperparameters of SNGP because ResNet has not been demonstrated directly. Considering the sensitivity to kernel hyperparameters, we consider the various length scales $l$ of the RBF kernel function. We train SNGP based on the experimental protocol in Appendix B.2.

Table 9 shows that SNGP achieves better AUROC for recognizing the OOD set compared to the proposed method. However, SNGP performs significantly worst on the IND set as comparing other baseline in Table 1.

| Model | Method | ACC ↑ | NLL ↓ | ECE ↓ | AUROC-S ↑ |
|---|---|---|---|---|---|
| | R-FVI (**our**) | (**0.952**, 0.001) | (**0.187**, 0.005) | (**0.028**, 0.001) | (0.956, 0.004) |
| ResNet 18 CIFAR 10 | SNGP ($l = 1 \times 10^0$) | (0.904, 0.013) | (0.395, 0.009) | (0.055, 0.005) | (0.993, 0.001) |
| | SNGP ($l = 5 \times 10^{-2}$) | (0.908, 0.005) | (0.423, 0.013) | (0.063, 0.002) | (0.993, 0.001) |
| | SNGP ($l = 1 \times 10^{-4}$) | (0.912, 0.005) | (0.412, 0.020) | (0.061, 0.003) | (**0.994**, 0.001) |
| | R-FVI (**our**) | (**0.799**, 0.003) | (**0.792**, 0.012) | (**0.056**, 0.002) | (0.850, 0.015) |
| ResNet 50 CIFAR 100 | SNGP ($l = 1 \times 10^0$) | (0.540, 0.017) | (1.957, 0.053) | (0.068, 0.016) | (0.953, 0.008) |
| | SNGP ($l = 5 \times 10^{-2}$) | (0.574, 0.023) | (2.242, 0.056) | (0.138, 0.025) | (**0.951**, 0.011) |
| | SNGP ($l = 1 \times 10^{-4}$) | (0.542, 0.015) | (2.220, 0.159) | (0.091, 0.051) | (0.927, 0.016) |

Table 9: Comparison R-FVI with SNGP on CIFAR-10 and CIFAR-100.

### B.2.4 COMPARISON WITH DEEP ENSEMBLE

We also compare the R-FVI with the Deep Ensemble (DE) Lakshminarayanan et al. (2017). As DE uses $n \times P$ parameters, where $P$ represents the number of single model parameters, and similarly requires $n \times T$ training time, where $T$ is the training time for a single model, we believe that comparing the DE version of R-FVI is fair as done in Rudner et al. (2022); Wilson & Izmailov (2020)

Thus, we compare DE, R-FVI, and Multi R-FVI (DE version of our method) using 5 member ensemble meaning one ensemble consists of 5 models trained independently. We report the results in Table 10.

| Model | Method | ACC ↑ | NLL ↓ | ECE ↓ | AUROC-S ↑ |
|---|---|---|---|---|---|
| ResNet 18 CIFAR 10 | R-FVI | (0.952, 0.001) | (0.162, 0.003) | (0.028, 0.001) | (0.956, 0.004) |
| | DE (5 member) | (0.961, 0.001) | (0.124, 0.002) | (0.007, 0.000) | (0.964, 0.007) |
| | Multi R-FVI (**our**) | (0.962, 0.001) | (0.123, 0.002) | (0.007, 0.000) | (0.963, 0.004) |
| ResNet 50 CIFAR 100 | R-FVI (**our**) | (0.799, 0.003) | (0.785, 0.013) | (0.056, 0.002) | (0.850, 0.015) |
| | DE (5 member) | (0.824, 0.003) | (0.654, 0.005) | (0.020, 0.001) | (0.848, 0.007) |
| | Multi R-FVI (**our**) | (0.824, 0.001) | (**0.644**, 0.005) | (0.020, 0.001) | (**0.860**, 0.005) |

Table 10: Comparison of R-FVI with DE and Multi-RFVI on CIFAR-10 and CIFAR-100.

### B.3 ADDITIONAL EXPERIMENT RESULTS FOR SECTION 5.3

**Experiment setting.** We basically follow the well-known training hyperparameters configurations in (Dosovitskiy et al., 2021). We use 128 batch size (4 step gradient accumulation with 32 batch size), and use 1000 steps for training PETS 37 dataset and 2000 steps for training DTD 47 dataset and AIRCRAFT 100 dataset ($T = 41$ epoch for PETS 37, $T = 77$ epoch for DTD 47, and $T = 43$ epoch for AIRCRAFT 100).

For optimizer, we use SGD optimizer with $1 \times 10^{-2}$ learning rate and 0.9 momentum. We use the cosine learning scheduler after consuming $0.1 \times$ total steps as warm-up steps. The other configuration of each inference method is described in Table 11.

| Inference | Hyperparameters | Range |
|---|---|---|
| MAP | Regularization $\lambda$ | $\{10^{-3}, 10^{-4}\}$ |
| T-FVI, R-FVI | KL regularization $\lambda$ in Eq. (1) | $\{10^{-5}, 10^{-6}\}$ |
| T-FVI, R-FVI | Variance of of variational weight parameters (log) | $\mathcal{U}(-6, -5)$ |
| T-FVI, R-FVI | The number of context inputs per batch | 32 / 128 (VIT) |
| R-FVI | Pre-determined iterations $\mathcal{T}$ | $\mathcal{T}_{\text{VIT}}$ |
| R-FVI | Radius $r$ in Eq. (13) for adversarial feature | $\{0.05, 0.10, 0.15\}$ |
| R-FVI | Scale of the variance of weight-space prior $\widehat{\sigma}_k^2$ | 10 |
| R-FVI | Restriction of function-space prior (TopK) | 5 (PETS 37 and DTD 47), 10 (AIRCRAFT 100) |

Table 11: Hyperparameters settings of the proposed inference (R-FVI)

For the R-FVI, we consider the following SGD trajectories $\mathcal{T}_{\text{VIT}} = \{\mathcal{T}_1, \mathcal{T}_2, \mathcal{T}_3, \mathcal{T}_4\}$ with $T$ epoch:

$\mathcal{T}_1 = \{0.5T - 10, 0.5T - 8, 0.5T - 6, 0.5T - 4, 0.5T - 2\}$,
$\mathcal{T}_2 = \{0.6T - 10, 0.6T - 8, 0.6T - 6, 0.6T - 4, 0.6T - 2\}$,
$\mathcal{T}_3 = \{0.7T - 10, 0.7T - 8, 0.7T - 6, 0.7T - 4, 0.7T - 2\}$,
$\mathcal{T}_4 = \{0.8T - 10, 0.8T - 8, 0.8T - 6, 0.8T - 4, 0.8T - 2\}$.

For computational resource, we used RTX 3090 TI (24 GB) to run experiments.

**Results for AIRCRAFT 100 dataset.** Table 12 shows the results of MAP, T-FVI, and R-FVI for the AIRCRAFT 100 dataset over 3 random seeds. We use $J$ predictive sample functions for Bayesian model averaging (BMA) prediction.

| SGD Trajectory | # sample | ACC ↑ | NLL ↓ | ECE ↓ | AUROC-S ↑ |
|---|---|---|---|---|---|
| MAP | J=1 | (0.701, 0.005) | (1.157, 0.008) | (0.094, 0.002) | (0.998, 0.001) |
| T-FVI | J=10 | (0.694, 0.000) | (1.255, 0.000) | (0.102, 0.000) | (0.998, 0.000) |
| | J=100 | (0.710, 0.000) | (1.166, 0.000) | (0.126, 0.000) | (0.999, 0.000) |
| R-FVI w. $\mathcal{T}_1$, $r = 0.10$ | J=10 | (0.706, 0.010) | (1.146, 0.031) | (0.033, 0.005) | (0.999, 0.000) |
| | J=100 | (**0.718**, 0.006) | (**1.060**, 0.027) | (**0.044**, 0.007) | (0.999, 0.000) |
| R-FVI w. $\mathcal{T}_2$, $r = 0.10$ | J=10 | (0.692, 0.009) | (1.201, 0.034) | (0.059, 0.006) | (0.998, 0.000) |
| | J=100 | (0.707, 0.007) | (1.114, 0.030) | (0.081, 0.006) | (0.999, 0.000) |

Table 12: Full results for AIRCRAFT 100 dataset

**Results of PETS 37.** Table 13 shows the results of MAP, T-FVI, and R-FVI for the PETS 37 dataset over 3 random seeds. We use $J$ predictive sample functions for Bayesian model averaging (BMA) prediction.

**Results of DTD 47 dataset.** Table 14 shows the results of MAP, T-FVI, and R-FVI for the DTD 47 dataset over 3 random seeds. We use $J$ predictive sample functions for Bayesian model averaging (BMA) prediction.

| SGD Trajectory | # sample | ACC ↑ | NLL ↓ | ECE ↓ | AUROC-S ↑ |
|---|---|---|---|---|---|
| MAP | J=1 | (0.940, 0.002) | (0.279, 0.005) | (0.038, 0.001) | (1.000, 0.000) |
| T-FVI | J=10 | (0.935, 0.001) | (0.245, 0.004) | (0.012, 0.001) | (1.000, 0.000) |
| | J=100 | (0.937, 0.001) | (0.223, 0.001) | (0.016, 0.002) | (1.000, 0.000) |
| R-FVI w. $\mathcal{T}_2$, $r = 0.10$ | J=10 | (0.941, 0.001) | (0.237, 0.002) | (0.016, 0.001) | (1.000, 0.000) |
| | J=100 | (0.942, 0.002) | (**0.213**, 0.003) | (0.012, 0.001) | (1.000, 0.000) |
| R-FVI w. $\mathcal{T}_3$, $r = 0.05$ | J=10 | (0.941, 0.003) | (0.236, 0.004) | (0.014, 0.002) | (1.000, 0.000) |
| | J=100 | (**0.942**, 0.001) | (**0.213**, 0.003) | (**0.009**, 0.001) | (1.000, 0.000) |
| R-FVI w. $\mathcal{T}_3$, $r = 0.10$ | J=10 | (0.942, 0.003) | (0.237, 0.002) | (0.016, 0.001) | (1.000, 0.000) |
| | J=100 | (**0.942**, 0.001) | (**0.213**, 0.002) | (0.010, 0.001) | (1.000, 0.000) |

Table 13: Full results for PETS 37 dataset

| SGD Trajectory | # sample | ACC ↑ | NLL ↓ | ECE ↓ | AUROC-S ↑ |
|---|---|---|---|---|---|
| MAP | J=1 | (0.790, 0.006) | (1.068, 0.016) | (0.131, 0.004) | (0.972, 0.004) |
| T-FVI | J=10 | (0.781, 0.010) | (0.906, 0.027) | (0.038, 0.003) | (0.983, 0.002) |
| | J=100 | (0.785, 0.009) | (0.801, 0.022) | (**0.029**, 0.004) | (0.988, 0.002) |
| R-FVI w. $\mathcal{T}_2$, $r = 0.10$ | J=10 | (0.784, 0.007) | (1.012, 0.073) | (0.076, 0.016) | (0.959, 0.031) |
| | J=100 | (0.790, 0.005) | (0.883, 0.06) | (0.065, 0.018) | (0.966, 0.029) |
| R-FVI w. $\mathcal{T}_3$, $r = 0.10$ | J=10 | (0.787, 0.004) | (0.900, 0.013) | (0.047, 0.003) | (0.982, 0.006) |
| | J=100 | (0.793, 0.001) | (**0.797**, 0.022) | (0.035, 0.004) | (**0.988**, 0.006) |
| R-FVI w. $\mathcal{T}_3$, $r \in \mathcal{U}(0.05, 0.15)$ | J=10 | (0.791, 0.002) | (0.927, 0.010) | (0.057, 0.002) | (0.980, 0.005) |
| | J=100 | (**0.794**, 0.000) | (0.817, 0.018) | (0.048, 0.002) | (0.986, 0.005) |
| R-FVI w. $\mathcal{T}_4$, $r = 0.10$ | J=10 | (0.783, 0.003) | (0.892, 0.014) | (0.040, 0.002) | (0.979, 0.006) |
| | J=100 | (0.790, 0.002) | (0.790, 0.020) | (0.032, 0.001) | (0.985, 0.005) |

Table 14: Full results for DTD 47 dataset

## B.4 ADDITIONAL EXPERIMENT RESULTS FOR SECTION 5.4

**Experiment settings.** Following the setting of UCI regression task in the appendix of Sun et al. (2019), we conduct the UCI regression task to demonstrate the effectiveness of the R-FVI. The baselines of the FVI Sun et al. (2019) and T-FVI Rudner et al. (2022) employ the GP prior with the RBF kernel and Neural Kernel Network (only for the protein set) as described in Sun et al. (2018). For the proposed method of R-FVI, we employ the hyperparameter described in Table 15. Then, we apply MAP inference for first 50 percent of the total training iterations to obtain the information from SGD trajectory, and then apply function-space variational inference for the remaining iterations.

| Hyperparameters | Range |
|---|---|
| learning rate | $\{10^{-3}, 10^{-4}, 3 \times 10^{-4}\}$, |
| KL regularization $\lambda$ in Eq. (1) | $\{0.1, 1.0\}$ |
| Variance of of variational weight parameters (log) | $\mathcal{U}(-6, -5)$ |
| The number of context inputs per batch | $(\#D_{\text{train}}/4)$ / $(\#D_{\text{train}})$ |
| Pre-determined iterations $\mathcal{T}$ | $\mathcal{T}_{\text{UCI}}$ |
| The number of sample functions $J$ | 100 |
| Radius $r$ in Eq. (13) for adversarial feature | $\{0.5, 1.0\}$ |
| Scale of the variance of weight-space prior $\widehat{\sigma}_k^2$ | 100 |

Table 15: Hyperparameters settings of the proposed inference (R-FVI)

We consider the SGD trajectory $\mathcal{T}_{\text{UCI}} = \{0.5T - 10, 0.5T - 8, 0.5T - 6, 0.5T - 4, 0.5T - 2\}$ with $T = 2000$ iterations and $T = 80000$ epochs (protein set).

For computational resource, we used RTX 4070 (12 GB) for UCI regression task.

**Additional results.** Fig. 17 describes the RMSE and Log likelihood (LL) over the UCI datasets.

(a) rmse ($\downarrow$ is better)

(b) log likelihood ($\uparrow$ is better)

Figure 17: RMSE and Log likelihood for UCI regression tasks

**Investigation on performance consistency over the different number of bins $K$.** We investigate on the consistency of the R-FVI performance as using the different number of the interval. Table 16 shows that R-FVI shows consistent performances across varying interval $K \in \{5, 10, 15\}$.

| Metric | Dataset | $K = 5$ | $K = 10$ | $K = 15$ |
|---|---|---|---|---|
| RMSE ($\downarrow$) | Boston | $2.521 \pm 0.371$ | $2.525 \pm 0.372$ | $2.530 \pm 0.375$ |
| | Concrete | $3.793 \pm 0.416$ | $3.777 \pm 0.466$ | $3.770 \pm 0.450$ |
| | Energy | $0.350 \pm 0.031$ | $0.349 \pm 0.036$ | $0.335 \pm 0.025$ |
| | Yacht | $0.422 \pm 0.119$ | $0.410 \pm 0.111$ | $0.410 \pm 0.115$ |
| | Wine | $0.510 \pm 0.026$ | $0.509 \pm 0.026$ | $0.509 \pm 0.026$ |
| | Protein | $3.611 \pm 0.039$ | $3.617 \pm 0.041$ | $3.617 \pm 0.041$ |
| Log likelihood ($\uparrow$) | Boston | $-1.806 \pm 0.202$ | $-1.808 \pm 0.197$ | $-1.810 \pm 0.200$ |
| | Concrete | $-2.464 \pm 0.293$ | $-2.509 \pm 0.339$ | $-2.575 \pm 0.479$ |
| | Energy | $0.255 \pm 0.147$ | $0.275 \pm 0.160$ | $0.307 \pm 0.127$ |
| | Yacht | $-0.530 \pm 1.053$ | $-0.528 \pm 1.085$ | $-0.512 \pm 1.148$ |
| | Wine | $-0.355 \pm 0.113$ | $-0.343 \pm 0.139$ | $-0.345 \pm 0.134$ |
| | Protein | $-2.018 \pm 0.008$ | $-2.019 \pm 0.009$ | $-2.019 \pm 0.009$ |

Table 16: RMSE and Log likelihood values for different number $K$ of interval.

