# OpenReview forum: "Adaptive Priors from Learning Trajectories for Function-Space Bayesian Neural Networks"
_ICLR.cc/2025/Conference — Submitted to ICLR 2025_

### Official Review · Reviewer_7Ju7 · 2024-11-01

**Soundness:** 3
**Presentation:** 3
**Contribution:** 2
**Rating:** 5
**Confidence:** 3

**Summary:**

In function-space VI there are two challenges: (1) how to specify meaning prior. (2) how to calculate the KL term in ELBO. This paper proposes to tackle the first challenge by first using SWAG to get a learning trajectory, and then constructing a tractable "prior" form based on it. Specifically, the authors only treat the last layer Bayesian and treat the feature extractor as fixed. The weighted sum of trajectory history is used to construct the feature extractor and the "prior". For the second challenge, the authors propose to use the adversarial context feature which removes the need of an external dataset for the context set. Experiment results show the proposed method has good uncertainty estimation in image classification and UCI regression.

**Strengths:**

- The adversarial context feature removes the need for an extra dataset as a context set.
- The constructed tractable "prior" is based on a posterior, and hence is capable of assign large variance for data differ from training set
- The paper is well-written and easy to follow.

**Weaknesses:**

- The proposed method is only for last-layer Bayesian neural network
- The construction of "prior" is computationally expensive and might be sensitive to the initialization

**Questions:**

- I put a quote sign on prior because this is more like a posterior. By definition prior is your belief in the problem before you see the data, but here the "prior" is constructed using SWAG and is loosely connected with the posterior obtained by SWAG. I think there needs to be at least a few sentences discussing this.

- The construction of the feature extractor seems overly complicated, is it really necessary to do so? What happens if you just use the MAP?

- The prior variance for T-FVI (line 472) seems pretty large (5, 10, 50), as I'm not very familiar with function space VI, I'm curious is this very common for FVI?

- As the adversarial context feature is combined with the tractable prior, it's hard to see the effect of it. How much does it affect the results when compared with using extra dataset as context set?

---

> ### Author Response · Authors · 2024-11-21
>
> ## Q1. Justification of the feature extractor.
>
> Our feature extractor allows the function-space prior to exhibit greater uncertainty when the input features are likely OOD. In contrast, directly employing the MAP solution cannot achieve this, as the function-space prior (linearized BNN) derived from the MAP solution cannot differentiate whether an input corresponds to IND features.
>
> As highlighted in the empirical results above (All reviewer), a comparison of the OOD performance between R-FVI and MOPED, which uses the MLE solution as the mean parameter of the weight-space prior, demonstrates that employing the MAP solution as the weight-space prior is not effective in enhancing OOD performance.
>
>
>
>
>
> ## Q2. Prior variance for T-FVI in line 472
>
> The prior variance for T-FVI is directly applied in function space (logits in the classification task). In our experiment (Table 1), each dimensional value of the logits of the trained ResNet ranges between $(-20, 20)$. Thus, we believe our choice of prior variance is reasonable.
>
> For setting the prior in FVI, regression tasks commonly use the trained Gaussian process regression. In classification tasks,  the prior variance is typically set as a hyperparameter, considering the range of the model's output.
>
>
>
>
> ## Q3. The effect of the an adversarial context feature
>
>
> We investigate the effect of the adversarial context feature $( z_{adv} $. To this end, we consider random noise perturbation $ z_{eps} = \widehat{h} + \epsilon $ with $ \epsilon \sim N(0, r^2I) $ instead of finding the adversarial context feature directly.
>
> As shown in Table 7 (Appendix, page 27), the adversarial context feature results in a better AUROC score, which denotes a measure of OOD uncertainty performance. Additionally, compared to R-FVI without function-space regularization, using the function-space KL regularization through our prior effectively improves ACC, NLL, ECE, and AUROC.

---

> > ### Comment · Reviewer_7Ju7 · 2024-11-25
> >
> > Thank you for answering my questions. However, I still have concerns regarding how the "prior" is constructed, SWAG has already seen the training data so it's not really a prior. Also I'm not entirely sure this is a fair comparison with other methods computational-wise. I'll keep my score as it is.

---

### Official Review · Reviewer_Lynm · 2024-11-01

**Soundness:** 2
**Presentation:** 1
**Contribution:** 3
**Rating:** 5
**Confidence:** 2

**Summary:**

The authors propose an explicit function-space prior for DNNs that can be applied to common model architectures and overcomes limitations of other function-space BNN approaches. They utilize DNNs as bayesian last-layer models and leverage the learning trajectory of the SGD from a set of checkpoints. They demonstrates the effectiveness across several (image) datasets and models, offering a practical method for function-space BNNs.

**Strengths:**

- **Clear structure**: From a macro perspective, the paper is well-organized, with a logical sequence of topics for good readability. Each section transitions smoothly, making it easy to follow the progression of ideas.
- **Contribution**: The authors seem to present an interesting and promising approach to function-space BNNs, addressing limitations that have previously hindered practical applications. The results demonstrate improvements over existing methods, showcasing the effectiveness of the proposed approach. However, since I am not very familiar with this field, I must trust the authors in this case.
- **Comprehensive introduction, limitations and related work**: The introduction effectively outlines the challenges of BNNs, particularly in the function-space domain, and establishes the motivation for this work. Section 3 is particularly well-structured, providing clear explanations of the limitations of BNNs and framing them within specific perspectives. The related work section makes clear distinctions between previous approaches and the authors' work.
- **Technical explanations**: The technical explanations seem well-organized, making the complex concepts easier to follow.
- **Diverse experimental perspectives**: The experiments cover three distinct perspectives, providing a quite comprehensive evaluation of the proposed approach and supporting its versatility.

**Weaknesses:**

**Grammar and Presentation**

Overall, the paper appears rushed and has numerous grammatical/wording issues. While they are minor individually, these errors accumulate and do not achieve the standard expected for an ICLR paper. Below is a list of specific issues I observed during my review (I eventually stopped listing them all):

- L.43, 46, 47-48, L.54: Improper usage or missing articles ("a," "the," pluralization issues).
- Abbreviations are repeatedly reintroduced (e.g., GP is introduced both in the intro and again at L.139).
- Abbreviation clarity: Some abbreviations have consistent capitalization (e.g., SGD), while ELBO only capitalizes "evidence".
- L.45: Should be "scalability issues" not "scalable issues".
- L.38: "NNs" is not introduced properly.
- L.39: "facilities" should be corrected to "facilitates".
- T-FVI is introduced for "function-space variational inference" but is inconsistently referenced (e.g., missing in L.54).
- L.409-410: "However, unlike theses work" is wrong.
- Inconsistent capitalization (e.g., L.432 has "Averaging" capitalized).
- References: VIT transformer paper is referenced twice; some arXiv papers are cited instead of their final venues.
In related work: many citations are not properly bracketed (using \citep) (e.g., L.401, 406, 417).


**Clarity**

Another major issue is the overall clarity, which makes the paper difficult to follow in detail:

- The contributions are not clearly stated, particularly with respect to differences in “…, and adaptively introduce different levels of uncertainty based on the function’s inputs” and “adaptively incorporate higher uncertainties for each function’s input”, which are used frequently. To me, it is particular unclear what “higher uncertainties” means and why/how it is useful. This terminology is used throughout the paper without sufficient explanation.

- "Higher-dimensional dataset" from the introduction needs clarification. What does this mean exactly, and why is it relevant to the discussion? This term is introduced for motivation but not addressed in the results. Are the datasets used in the results actually high-dimensional?

- The introduction would benefit from a brief explanation of why BNNs are advantageous in the first place.

- The phrase "widely-used DNN architectures" could be more precise. Does the method apply to all CNNs, transformers, or specific models? Also, clarify whether it applies to image classification models exclusively or to other classification tasks as well (only mentioned shortly in the abstract). Are the methods also applicable to other domains? I do not suggest do to more experiments but rather be more precise about the domain and maybe mention limitations/possibilities in others. The data types should be more clearly stated in the introduction.


**Result presentation**

The results section is short and difficult to follow due to the extensive usage of the appendix. It may be more effective to focus on two of the perspectives or to create more space for a comprehensive discussion.

- Table 1: The abbreviations for methods are not introduced. Although explained in the appendix, the table itself is hard to interpret without this.
- Tables and figures in general not well-placed throughout the text (e.g., Figure 5 is referenced before Figure 4, and Figure 4 appears two pages before it is discussed). This disrupts the flow and makes it challenging for the reader.
- While the design decisions regarding datasets and models may be understandable for image domain experts, a brief motivation for these choices would be beneficial.
- It’s unclear why only a subset of methods was selected for Table 2 (transfer approach); why?
- Table 2: Metrics like auroc-c and auroc-s are not properly introduced in the text.
- The sections on transfer learning and regression are very brief, which is especially notable as the transfer approach is a highlighted contribution. Metrics and abbreviations for the regression task are not well-introduced, causing further confusion..

**Short Conclusion**
- The conclusion is very short and lacks a discussion of future work.
- While it’s good that limitations are noted, the impact on practical adoption should be discussed. Additionally, suggestions on how this might be improved in the future would add value.

**Questions:**

I encourage the authors to address the weaknesses highlighted above, as well as to provide answers to the questions.

---

> ### Author Response · Authors · 2024-11-21
>
> Thank you for your suggestions. We will reflect your comments into the revision of our manuscript.

---

> > ### Comment · Reviewer_Lynm · 2024-11-22
> >
> > Dear authors, could you please provide details about the changes or issues you plan to address? Additionally, I would appreciate it if you could revisit and respond to the questions I raised.

---

> > > ### Author Response · Authors · 2024-11-23
> > >
> > > Regarding the possible revision plan in this rebuttal, we plan to move the explanation of the baselines, currently described in the appendix, into the main text and revise the sentence you suggested. Additionally, we respond to the following questions:
> > >
> > >
> > > ### Comment on statement for contribution
> > >
> > > The parameters of Bayesian models are random variables. The prior distribution over these parameters, often referred to as the weight-space prior, is designed to incorporate prior knowledge into the model and facilitate the derivation of the posterior distribution. However, defining an appropriate weight-space prior for a Bayesian neural network (BNN) is challenging, as we lack a clear understanding of how the chosen prior impacts the network's output. As a result, general BNNs typically adopt a zero-mean Gaussian prior by default.
> > >
> > > In contrast, when considering a BNN in function space, a more intuitive form of prior knowledge can be incorporated. Specifically, if an input is likely to belong to an out-of-distribution (OOD) dataset, the model's output should reflect higher uncertainty (i.e., high variance), signaling a lack of confidence in predictions for such inputs. We claim that our work introduces a function-space prior that effectively incorporates this bias.
> > >
> > > ----
> > >
> > > ### Comment on High dimensional dataset
> > >
> > > We evaluate our method primarily on image datasets $(D = 3 \times 32 \times 32$ for Table 1 and $D = 3 \times 224 \times 224$ for Table 2). These dataset dimensions are significantly larger than those typically used for regression or classification tasks $D \leq 10$ in general), where Gaussian processes (GPs) are commonly employed for function-space priors.
> > >
> > > ----
> > >
> > > ### Comment on Widely-used DNN architectures
> > >
> > > Our function-space prior is designed under the assumption that a DNN consists of a feature extractor and a last-layer classifier, which represents a common model structure. Therefore, our prior can be constructed for any model, regardless of its components (e.g., MLP layers, CNN layers, or attention layers), as long as the model adheres to this structure. Indeed, in our experiments, we have constructed the prior for ResNet in image classification, Vision Transformer in transfer learning, and MLP layers in regression tasks.
> > >
> > > ### Question on Table 2.
> > >
> > > The experiment using the ViT model requires more GPU memory because the ViT model used in our experiment is approximately four times larger than ResNet-50. As a result, we could not run all the baselines listed in Table 1 and instead selected a subset of baselines that demonstrated strong performance in Table 1.

---

> > > > ### Comment · Reviewer_Lynm · 2024-11-25
> > > >
> > > > Thank you for addressing the questions and providing clarifications. E.g., it would be helpful to explicitly include the reasoning behind the selection of only a subset of models in Table 2, as your response effectively answers this. Considering the concerns raised by other reviewers, my limited familiarity with the field, and the ongoing doubts regarding the contributions, I'll maintain my current score.

---

### Official Review · Reviewer_i5t7 · 2024-11-05

**Soundness:** 3
**Presentation:** 3
**Contribution:** 2
**Rating:** 5
**Confidence:** 3

**Summary:**

This paper proposes a novel method for specifying function-space priors in Bayesian neural networks that can be integrated to widely used neural network architectures. The parameters of these explicit functions are determined using the weight statistics over the learning trajectory by leveraging stochastic weight averaging Gaussian (SWAG) method. The work is motivated to address the scalability challenges of function-space priors such as Gaussian process prior in large neural network architectures with many parameters and high dimensional datasets. The proposed approach utilize context feature to compute the function-space KL divergence without relying on external datasets. Empirical evaluation is performed with experiments on image classification and regression, demonstrating the proposed method is effective in improving uncertainty estimation.

**Strengths:**

* The paper addresses an important problem of specifying meaningful function-space prior in Bayesian neural networks based on the learning trajectories through data.

* The use of adversarial context feature to compute the functional-space KL-divergence without relying on external dataset is an interesting approach, which could make the model to be more robust.

*  Empirical results supports the proposed method. However, the empirical evaluation could be strengthened by including experiments where all layers of the model are Bayesian. Please see the comments in the next section.

**Weaknesses:**

* While the paper is motivated to address the scalability issues of traditional Gaussian process priors with high-dimensional datasets and large neural network architectures with many parameters, the experimental evaluation is limited to a model where only the final layer is Bayesian. This raises concerns about the scalability of the proposed approach. Additionally, prior works [Ovadia et al. 2019, Xiao et al. 2022] has shown that the quality of uncertainty estimation in models where only the last layer is Bayesian is inferior to other comparable methods, such as Deep Ensembles and Monte Carlo Dropout. I encourage the authors to evaluate with a model (e.g. ResNet-18), where all layers are Bayesian in their experiments. This analysis would address the scalability concerns of the proposed function-space priors.

* The paper provides literature review of related works, but do not compare with any of them in their experiments. For example, how does the proposed method compare with other methods that define priors through Empirical Bayes [Krishnan et al. 2020, Shwartz-Ziv et al. 2022]

[Ovadia et al. 2019] Ovadia, Yaniv, Emily Fertig, Jie Ren, Zachary Nado, David Sculley, Sebastian Nowozin, Joshua Dillon, Balaji Lakshminarayanan, and Jasper Snoek. "Can you trust your model's uncertainty? evaluating predictive uncertainty under dataset shift." Advances in neural information processing systems 32 (2019).

[Xiao et al. 2022] Xiao, Yuxin, Paul Pu Liang, Umang Bhatt, Willie Neiswanger, Ruslan Salakhutdinov, and Louis-Philippe Morency. "Uncertainty Quantification with Pre-trained Language Models: A Large-Scale Empirical Analysis." In Findings of the Association for Computational Linguistics: EMNLP 2022, pp. 7273-7284. 2022.

[Krishnan et al. 2020] Krishnan, Ranganath, Mahesh Subedar, and Omesh Tickoo. "Specifying weight priors in bayesian deep neural networks with empirical bayes." In Proceedings of the AAAI conference on artificial intelligence, vol. 34, no. 04, pp. 4477-4484. 2020.

[Shwartz-Ziv et al. 2022] Shwartz-Ziv, Ravid, Micah Goldblum, Hossein Souri, Sanyam Kapoor, Chen Zhu, Yann LeCun, and Andrew G. Wilson. "Pre-train your loss: Easy bayesian transfer learning with informative priors." Advances in Neural Information Processing Systems 35 (2022): 27706-27715.

**Questions:**

* what is the motivation behind the adversarial context feature, specifically why FGSM attack method? Any other perturbations/corruptions to the features will serve the purpose of not relying on the external dataset?

---

> ### Author Response · Authors · 2024-11-21
>
> ## Q1. Comparison with Empirical Bayes
>
> We agree that the empirical Bayes [Krishnan et al. 2020, Shwartz-Ziv et al. 2022] should be considered. Thus, we compare our approach in setting of Table 1 and 2. Further details, please see our reply for all reviewers above.
>
>
>
> ## Q2. Motivation behind the adversarial context feature
>
> Our adversarial context feature is defined as the feature $h$ in the vicinity of $\widehat{h}(x)$ which has the largest MHD distance between $h$ and its nearest IND feature $m_{q_x}$. Intuitively, as shown in Figure 2, $z_{adv}$ is set by pushing $\widehat{h}_2 (x) $ further way from $m_1$ so that the value of its function-space variance increases. To this end, we solve an optimization problem similar to the adversarial attack, where the perturbed input $\tilde{x}$ is found  in the vicinity of $x$ that maximizes the training loss of $x$.

---

> > ### Comment · Reviewer_i5t7 · 2024-11-22
> >
> > Thank you for your response and additional experiments. Can you provide clarification on the experiments in Table 1 comparing with MOPED :
> >
> > (1) are you applying your method to only the final layer of ResNet, or all the parameters of the model?  This is related to my earlier comment - *" experimental evaluation is limited to a model where only the final layer is Bayesian. This raises concerns about the scalability of the proposed approach. Additionally, prior works [Ovadia et al. 2019, Xiao et al. 2022] has shown that the quality of uncertainty estimation in models where only the last layer is Bayesian is inferior to other comparable methods, such as Deep Ensembles and Monte Carlo Dropout"*
> >
> > (2) which dataset?

---

> > > ### Author Response · Authors · 2024-11-23
> > >
> > > For (1), we apply our method only to the final layer of ResNet. This is because constructing the Jacobian matrix for fully Bayesian layers in ResNet poses GPU memory issues, as discussed in Section 3 (Limitations), making it infeasible.
> > >
> > >
> > > For (2), we use the CIFAR-10 and CIFAR-100 datasets, which are the same datasets used in Table 1.

---

> > > > ### Comment · Reviewer_i5t7 · 2024-11-23
> > > >
> > > > Thank you for the clarification. However, it does not address the primary concern I had regarding the scalability of the propose method. This limitation, in turn, prevents a fair comparison with the existing methods, as mentioned in my earlier comments. - *"This raises concerns about the scalability of the proposed approach. Additionally, prior works [Ovadia et al. 2019, Xiao et al. 2022] has shown that the quality of uncertainty estimation in models where only the last layer is Bayesian is inferior to other comparable methods, such as Deep Ensembles and Monte Carlo Dropout"*. I have updated my assessment.

---

### Official Review · Reviewer_rfj3 · 2024-11-08

**Soundness:** 1
**Presentation:** 2
**Contribution:** 1
**Rating:** 1
**Confidence:** 3

**Summary:**

The paper introduces a new prior designed for functional variational inference in Bayesian neural networks. To achieve this, the authors propose an explicit function-space prior by modeling deep neural networks (DNNs) as Bayesian last-layer models. Initially, the method constructs a prior in weight space similar to the approach used in SWAG (stochastic weight averaging Gaussian). From this weight-space prior, the corresponding functional prior is derived. The effectiveness of the proposed method is demonstrated through validation on tasks including image classification, transfer learning, and UCI regression.

**Strengths:**

- The paper aims to tackle an important problem which is choosing sensible priors for Bayesian neural networks.
- The paper is well-written and provide good illustrative figures that enhance the clarity of method.

**Weaknesses:**

-	The contribution is marginal, as functional variational inference is a well-established area in the literature. Additionally, the approach of constructing priors using SWAG has already been explored by Shwartz-Ziv et al. (2022).
-	The proposed prior is not valid from a Bayesian perspective, as it resembles a posterior. It is constructed from the training data and multiple trained models, as illustrated in Figures 2 and 5.
-	The idea presented in this paper appears to be very similar to the work found at https://openreview.net/pdf?id=AZVmYg3LvS. If this paper is a revised version, a comparison with Table 2 in that paper raises a concern: why are all the results for the baseline methods in Table 1 significantly worse, particularly for the NLL, ECE, and AUROC metrics?
-	From lines 47 to 54, the paper discusses the work of Rudner et al. (2022), highlighting that it restricts the randomness to the last layer due to GPU memory limitations, which can reduce the flexibility of Bayesian neural networks (BNNs). However, the paper ultimately adopts a similar approach, effectively presenting a simple extension of that work. This could be misleading to readers, as the paper suggests a novel contribution while largely replicating the limitations discussed.

**Questions:**

Please see the third bullet in the Weaknesses

---

> ### Author Response · Authors · 2024-11-21
>
> ## Q1. Marginal contribution as comparing to Shwartz-Ziv et al. (2022)
>
> We would like to emphasize that our function-space prior adaptively incorporates higher uncertainty into the function’s output when an input feature is more likely to be an in-distribution (IND) feature, as stated in Proposition 4.1 and demonstrated in Figure 5 and the Appendix (pages 22 and 24). We believe this approach is clearly distinct from [1], which sets the weight-space prior using SWAG but does not consider the effect of the prior in function space, i.e., it does not consider how this imposed weight-space prior influences the function’s output.
>
> [1] Pre-Train Your Loss: Easy Bayesian Transfer Learning with Informative Priors - NeurIPS 22
>
>
>
> ## Q2. Justification of the proposed prior in Bayesian perspective.
>
> We partially agree that the proposed prior may not be valid from a Bayesian perspective. However, our prior construction can be regarded as an empirical Bayesian approach, where the parameters of the prior are set based on the training dataset. Indeed, [1] and [2] also adopt empirical Bayesian approaches: [1] trains on upstream datasets for transfer learning, and [2] uses pre-trained parameters obtained from learning on the same dataset.
>
> [1] Pre-Train Your Loss: Easy Bayesian Transfer Learning with Informative Priors - NeurIPS 22
>
> [2] Specifying weight priors in Bayesian deep neural networks with empirical bayes - AAAI 20
>
>
>
>
> ## Q3. Question of the baseline performances regarding the resubmission.
>
> This is a re-submission of previous work available at https://openreview.net/pdf?id=AZVmYg3LvS. The reason all baseline performances (NLL, ECE, and AUROC) in Table 1 are significantly worse is that Table 1 reports the calibrated versions of NLL, ECE, and AUROC based on [1, 2], which are known to be better than the non-calibrated scores. Notably, the performance of the proposed method also degrades when compared to the results in Table 1.
>
> [1] Pitfalls of in-domain uncertainty estimation and ensembling in deep learning - ICLR 20
>
> [2] https://github.com/SamsungLabs/pytorch-ensembles/blob/metrics.py

---

> ### Comment · Reviewer_rfj3 · 2024-11-22
> **Official Comment by Reviewer rfj3**
>
> Thanks the authors for providing clarifications.
>
> However, the reviewer continues to have significant doubts regarding the baseline performances compared to the earlier version of the paper. Neither version discusses or cites the work on calibrated scores.
>
> This is a critical issue, as the reported results must be accurate and well-justified.
>
> As a result, I would maintain my current assessment.

---

### Official Review · Reviewer_Z8Gp · 2024-11-11

**Soundness:** 2
**Presentation:** 3
**Contribution:** 2
**Rating:** 3
**Confidence:** 4

**Summary:**

This manuscript presents a function-space prior that can be integrated into common deep neural network architectures, treating DNNs as Bayesian last-layer models. The mean and variance functions for the prior parameters are set using weight and feature statistics from the learning trajectory.

**Strengths:**

The manuscript focuses on the important aspect of using informative priors and applying them to efficient Bayesian inference.

**Weaknesses:**

The important weakness of the work is limited novelty. The method combines the two approaches SWAG and T-FSVI. The extensions proposed in this work are limited in novelty. There are already some works (eg. [1]) in the literature that capture the weight priors from the phase-I of training the neural network models which provide better accuracy and uncertainty quantification, for larger models.
From the results, the benefits of phase-I are limited.

[1] Krishnan, Ranganath, et.al. "Specifying weight priors in Bayesian deep neural networks with empirical bayes." Proceedings of the AAAI conference on artificial intelligence. Vol. 34. No. 04. 2020.

**Questions:**

Please refer to comments in weakness section.

---

> ### Author Response · Authors · 2024-11-21
>
> ## Q1. Comment for incremental novelty.
>
> We partially agree that our prior shares some similarities with [1], because both approaches employ the learning trajectory (i.e., phase I) to specify the mean parameters of the prior.
>
> However, [1] proposed setting the mean parameter of a weight-space prior as MLE parameter obtained from learning trajectory. [1] does not specify how to set the variance parameters of prior and just sets it as the hyperparameter.
>
> On the other hand, our work focuses on building a function-space prior unlike [1] focusing on the weight-space prior. In addition, our approach allows the function-space prior to have larger uncertainty depending on whether the feature of an input is likely to be an in-distribution (IND) feature, as stated in Proposition 4.1.
>
> We believe this is a novel contribution that differentiates our work from [1]. Additionally, to the best of our knowledge, using the learning trajectory within a function-space prior to adaptively incorporate uncertainty has not been explored before.
>
> [1] Specifying weight priors in Bayesian deep neural networks with empirical bayes - AAAI 20

---

> > ### Comment · Reviewer_Z8Gp · 2024-12-03
> >
> > I thank the authors for providing the feedback. After reviewing all the comments, I'll keep my current rating.

---

### Author Response · Authors · 2024-11-21
**To all reviewers**

## To all reviewers

Thank you for reviewing our work. Based on the feedback from the reviewers, it seems that our function-space prior construction is viewed to have an incremental novelty, and that an empirical comparison with empirical Bayes should be included. However, we believe that our prior construction is clearly distinct from the mentioned weight-space prior construction, and thus have a clear novelty. We would like to clarify our contribution. Additionally, we will present an empirical comparison with empirical Bayes approaches.


----

## Comment on Incremental novelty

One of our main objective is to build a function-space prior that can incorporate varying levels of uncertainty, depending on whether an input is likely to belong to the IND set or not as Gaussian process prior does. To achieve this, we use the statistics of the last-layer features and weight parameters, obtained from SWAG, to construct the function-space prior. Indeed, we demonstrate that our function-space prior introduces greater uncertainty into the function's output when the feature of an input is likely to be from the OOD set, as stated in Proposition 4.1. This property cannot be achieved by directly applying SWAG to construct the function-space prior. We believe this is a novel contribution that differentiates our work from the previous works of [1] and [2].

Additionally, although the previous work [1] shares some similarities with our approach, particularly in using trained parameters for the weight-space prior (i.e., the empirical Bayes approach), [1] employs the pre-trained parameters as the mean of the weight-space prior without elaborating on how to specify the variance parameters of the weight-space prior.

[1] Specifying weight priors in Bayesian deep neural networks with empirical bayes - AAAI 2020

[2] Pre-train your loss: Easy bayesian transfer learning with informative priors - NeurIPS 2022


## Empirical comparison with Empirical Bayes

We compare our work with Empirical Bayesian approach under the same setting of the Image classification task (Table 1) and Transfer learning task (Table 2).

**To compare with the MOPED approach from [1]**, we train the ResNet models for the epochs $\\{T \times 0.75 ,T \times 0.80  ,T \times 0.85 \\}$, which are similar to the pre-trained epochs of R-FVI, and obtain $\widehat{\theta}_{mle}$.

Next, we set the weight-space prior to $N ( \widehat{\theta}_{mle}, \sigma^{2} )$ with $\sigma \in \\{0.1,1\\}$ and apply variational inference (VI) for the remaining training iterations (up to $T$ epochs).

The table below shows that our approach (R-FVI) outperforms MOPED. Additionally, since the performance of MOPED is highly sensitive to when $ \widehat{\theta}_{mle} $ is obtained, we believe that this highlights the importance of averaging the weight samples, obtained from the trajectory, for the prior construction.


| Dataset   | Model               | ACC | NLL   | ECE  | AUROC   |
|-----------|----------------------|-------------------|-------------------|-------------------|-----------------------|
| **CIFAR-10**  | MOPED (nep=$T \times 0.75$)    | (0.904, 0.009)    | (0.303, 0.031)    | (0.026, 0.003)    | (0.907, 0.028)        |
|           | MOPED (nep=$T \times 0.80$)    | (0.921, 0.002)    | (0.271, 0.008)    | (0.032, 0.009)    | (0.929, 0.014)    |
|           | MOPED (nep=$T \times 0.85$)    | (0.943, 0.000)    | (0.219, 0.005)    | (0.032, 0.000)    | (0.921, 0.018)        |
|           | R-FVI               | (**0.952**, 0.001) | (**0.187**, 0.005) | (0.028, 0.001)    | (**0.956**, 0.004)        |
| **CIFAR-100** | MOPED  (nep=$T \times 0.75$)     | (0.705, 0.012)    | (1.136, 0.046)      | (0.069, 0.007)    | (0.819, 0.033)    |
|           | MOPED  (nep=$T \times 0.80$)   | (0.721, 0.004)    | (1.136, 0.029)      | (0.086, 0.020)    | (0.806, 0.030)        |
|           | MOPED  (nep=$T \times 0.85$)    | (0.790, 0.002)    | (0.864, 0.009)      | (0.078, 0.000)    | (0.791, 0.008)        |
|           | R-FVI               | (**0.799**, 0.003) | (**0.792**, 0.012) | (**0.056**, 0.001)    | (**0.850**, 0.015)        |

---

> ### Author Response · Authors · 2024-11-21
>
> **To compare with the PTYL approach from [2]**, we construct the SWAG weight-space prior (rank=10) by further training the pre-trained VIT-B16 on ImageNet 1k. Then, we finetune the pre-trained VIT-B16 using CSGHMC of [3] with SWAG weight-space prior.
>
> The table below shows that our approach (R-FVI) outperforms PTYL in terms of accuracy and underperforms in terms of ECE
>
>
> | Dataset                 | Model   | ACC  | NLL  | ECE  | AUROC-S | AUROC-C |
> |---------------|---------|-------------------|-------------------|-------------------|---------------------|---------------------|
> | **PETS 37**             |  PTYL  | (0.942, 0.002)    | (**0.192**, 0.005)    | (0.019, 0.002)    | (1.000, 0.000)       | (0.999, 0.000)    |
> |                                  | R-FVI   | (0.942, 0.002)    | (0.215, 0.003)    | (**0.010**, 0.001) | (1.000, 0.000)       | (0.999, 0.000)    |
> |                                  | T-FVI   | (0.937, 0.002)    | (0.225, 0.003)    | (0.015, 0.001) | (1.000, 0.000)       | (0.999, 0.000)    |
> | **DTD 47**              |  PTYL  | (0.781, 0.003)    | (0.780, 0.007) | (**0.021**, 0.002)    | (0.992, 0.003)      | (0.982, 0.002)      |
> |                                 | R-FVI   | (**0.793**, 0.002) | (0.795, 0.022)    | (0.033, 0.004)    | (0.988, 0.006)      | (0.983, 0.001)      |
> |                                  | T-FVI   | (0.785, 0.009)    | (0.801, 0.022)    | (0.029, 0.004)      | (0.988, 0.002)       | (0.985, 0.004)    |
> | **AIRCRAFT 100**        |  PTYL  | (0.695, 0.008)    | (1.042, 0.007) | (**0.028**, 0.001) | (0.999, 0.001)      | (0.996, 0.002)      |
> |                                       | R-FVI   | (**0.718**, 0.006) | (1.055, 0.033)    | (0.045, 0.008)    | (0.999, 0.000)      | (0.998, 0.001)      |
> |                                        | T-FVI   | (0.711, 0.000)    | (1.155, 0.001)    | (0.124, 0.000) | (0.999, 0.000)       | (0.998, 0.000)    |
>
> [1] Specifying weight priors in Bayesian deep neural networks with empirical bayes - AAAI 2020
>
> [2] Pre-train your loss: Easy bayesian transfer learning with informative priors - NeurIPS 2022
>
> [3] Cyclical Stochastic Gradient MCMC for Bayesian Deep Learning  - ICLR 2020

---

### Meta-Review · Area_Chair_pdM9 · 2024-12-20

**Metareview:**

This paper proposes a novel approach to specifying function-space priors in Bayesian neural networks, leveraging the learning trajectory of stochastic gradient descent to construct a tractable prior form. The authors demonstrate the effectiveness of their method in image classification and UCI regression tasks, showcasing improved uncertainty estimation.

The reviewers generally agree that the paper addresses an important problem in Bayesian neural networks, but raise several concerns regarding the novelty, soundness, and presentation of the work. Some reviewers question the marginal contribution of the paper, citing similarities with existing approaches such as SWAG and empirical Bayes. Others point out limitations in the experimental evaluation, including the restriction to last-layer Bayesian neural networks and the lack of comparison with other methods.

The reviewers also highlight issues with the presentation, including grammatical errors, inconsistent notation, and unclear explanations of key concepts. Additionally, some reviewers express concerns about the construction of the prior, which is based on a posterior distribution obtained through SWAG, and the potential sensitivity of the method to initialization.

After discussion with the authors and among themselves, all reviewers agree to reject the paper in its current form. We would still like to encourage the authors to resubmit an improved version of the paper in the future.

**Additional Comments On Reviewer Discussion:**

see above

---

### Decision · Program_Chairs · 2025-01-22

Reject